# Differences in topological progression profile among neurodegenerative diseases from imaging data

Sara Garbarino[1,2]*, Marco Lorenzi[2], Neil P Oxtoby[1], Elisabeth J Vinke[3], Razvan V Marinescu[1], Arman Eshaghi[1,4], M Arfan Ikram[3,5], Wiro J Niessen[5], Olga Ciccarelli[4], Frederik Barkhof[1,6], Jonathan M Schott[7], Meike W Vernooij[3,5], Daniel C Alexander[1], for the Alzheimer's Disease Neuroimaging Initiative

[1]Centre for Medical Image Computing, Department of Computer Science, University College London, London, United Kingdom; [2]Université Côte d'Azur, Inria, Epione Research Project, Sophia Antipolis, France; [3]Department of Epidemiology, Erasmus Medical Center, Rotterdam, Netherlands; [4]Queen Square Multiple Sclerosis Centre, UCL Queen Square Institute of Neurology, Faculty of Brain Sciences, University College London, London, United Kingdom; [5]Department of Radiology and Nuclear medicine, Erasmus MC, Rotterdam, Netherlands; [6]Department of Radiology and Nuclear medicine, VUmc, Amsterdam, Netherlands; [7]Dementia Research Centre, Institute of Neurology, University College London, London, United Kingdom

**Abstract** The spatial distribution of atrophy in neurodegenerative diseases suggests that brain connectivity mediates disease propagation. Different descriptors of the connectivity graph potentially relate to different underlying mechanisms of propagation. Previous approaches for evaluating the influence of connectivity on neurodegeneration consider each descriptor in isolation and match predictions against late-stage atrophy patterns. We introduce the notion of a *topological profile* — a characteristic combination of topological descriptors that best describes the propagation of pathology in a particular disease. By drawing on recent advances in disease progression modeling, we estimate topological profiles from the full course of pathology accumulation, at both cohort and individual levels. Experimental results comparing topological profiles for Alzheimer's disease, multiple sclerosis and normal ageing show that topological profiles explain the observed data better than single descriptors. Within each condition, most individual profiles cluster around the cohort-level profile, and individuals whose profiles align more closely with other cohort-level profiles show features of that cohort. The cohort-level profiles suggest new insights into the biological mechanisms underlying pathology propagation in each disease.

*For correspondence:
sara.garbarino@inria.fr

## Introduction

Evidence from neuroimaging suggests that the progression of brain changes in neurodegenerative diseases may be mediated by brain connectivity. For example, atrophy patterns observed with MRI suggest that different brain regions are systematically and selectively vulnerable to different neurodegenerative diseases, and that these atrophy patterns closely match known connectivity networks (*Iturria-Medina, 2013*; *Iturria-Medina and Evans, 2015*; *Raj et al., 2012*; *Seeley et al., 2009*; *Zhou et al., 2012*). The literature includes wide debate on the potential mechanisms underlying pathogenic protein propagation, on the biology of protein aggregation and propagation, and of selective vulnerability of neurons in neurodegenerative disease (see *Soto and Pritzkow, 2018*;

*Jucker and Walker, 2018*; *Fu et al., 2018* for recent reviews). Uncovering the precise relationship between the topology of brain network connectivity and the pattern of pathology may provide clues to the underlying mechanisms of propagation. Indeed, in Alzheimer's disease (AD), *Zhou et al. (2012)* attempt to explain the association between patterns of brain atrophy and connectivity in terms of different topological descriptors: (1) *centrality* – the most active regions or 'hubs' are most vulnerable (*Buckner et al., 2009*; *Saxena and Caroni, 2011*); (2) *segregation* – opposite to 1), the most isolated nodes are most vulnerable (*Appel, 1981*; *Salehi et al., 2006*); (3) *network proximity* – regions connected to disease epicenters are most vulnerable (*Frost and Diamond, 2010*; *Jucker and Walker, 2013*; *Prusiner, 1984*); (4) *cortical proximity* – regions spatially adjacent to disease epicenters are most vulnerable. The authors relate these descriptors to distinct mechanisms of pathology propagation: (1) centrality represents nodal stress, (2) segregation, trophic failure, (3) network proximity, trans-neuronal spread, and (4) cortical proximity represents unguided diffusive propagation. Thus, comparing patterns of pathology predicted by these different descriptors with those observed in patient cohorts gives clues to which corresponding mechanisms are at play. Moreover, as reviews (*Soto and Pritzkow, 2018*; *Jucker and Walker, 2018*; *Fu et al., 2018*) highlight, the plausibility of those mechanisms extend similarly to the wider range of neurodegenerative conditions.

Previous studies (*Iturria-Medina, 2013*; *Iturria-Medina and Evans, 2015*; *Seeley et al., 2009*; *Zhou et al., 2012*; *Saxena and Caroni, 2011*; *Buckner, 2005*; *Cope et al., 2018*; *Fornito et al., 2015*; *Mancini, 2016*) using functional or structural networks have focused on evaluating the ability of each individual topological descriptor above to explain observed patient data with the goal of identifying the single most likely mechanism of disease propagation. More generally, a variety of mathematical models based on dynamical systems modeling (*Raj et al., 2015*; *Raj et al., 2012*; *Iaccarino et al., 2018*; *Weickenmeier et al., 2019*; *Mišić et al., 2015*; *Iturria-Medina et al., 2017*; *Iturria-Medina et al., 2018*) have been proposed for describing the temporal propagation of pathology mediated by brain networks. They mostly enforce the trans-neuronal spread (or 'prion-like') hypothesis of propagation via network proximity. These approaches have important methodological limitations, which should prompt caution in their interpretation. Two key limitations are: i) the approaches do not consider the possibility of multiple concurrent spreading mechanisms; ii) the evaluation of topological descriptors is performed using only cross-sectional data assumed to represent end-stage atrophy patterns.

Regarding limitation (i), multiple distinct mechanisms are likely to contribute in diseases where multiple proteinopathies are at play (such as amyloid and tau in AD; see for example *Iaccarino et al., 2018*; *Jones et al., 2016*; *Leal et al., 2018*); in diseases exhibiting highly-variable atrophy patterns (as in multiple sclerosis (MS); see *Steenwijk et al., 2016*); or in individual cases where multiple pathologies co-exist, as for instance tau accumulation in Parkinson's disease (*Irwin et al., 2013*; *Lei et al., 2010*), or alpha-synuclein or TDP-43 in AD (*Arai et al., 2009*; *Hamilton, 2006*; *Spires-Jones et al., 2017*; *Attems and Jellinger, 2014*). Also, associations between vascular factors and neurodegenerative dementias such as AD are common (*Kalaria, 2009*; *Sweeney et al., 2018*), which suggests contributions from multiple underlying mechanisms to the observed pattern of pathology accrual (*Attems and Jellinger, 2014*). Further, in MS, retrograde neurodegeneration secondary to focal damage from remote lesions, iron accumulation in the deep gray matter and 'virtual hypoxia' in the hub regions could all potentially combine and contribute to explain observed neurodegeneration (*Trapp and Stys, 2009*).

Regarding limitation (ii), considering only end-stage pathology severely limits sensitivity — like trying to guess the plot of a movie after watching only the final scene (*Oxtoby and Alexander, 2017*). The emergence of data-driven disease progression models (DPMs) (*Oxtoby and Alexander, 2017*; *Bilgel et al., 2016*; *Donohue et al., 2014*; *Eshaghi et al., 2018a*; *Firth et al., 2016*; *Fonteijn et al., 2012*; *Iturria-Medina et al., 2016*; *Jedynak et al., 2012*; *Koval et al., 2017*; *Li et al., 2019*; *Marinescu et al., 2019*; *Oxtoby et al., 2017*; *Oxtoby et al., 2018*; *Schiratti et al., 2017*; *Venkatraghavan et al., 2017*; *Young et al., 2014*) provides an opportunity to address limitation (ii) by using the full trajectory of pathology accumulation to evaluate the influence of topological descriptors. These techniques estimate the long-term temporal pattern of disease progression directly from cross-sectional or short-term longitudinal data sets on the assumption of some degree of commonality of progression over a patient cohort. In contrast to brute-force machine-learning approaches (*Hinrichs et al., 2011*; *Moradi et al., 2015*; *Nie et al., 2017*) to predict progression,

DPMs reveal the trajectory of temporal evolution of multiple biomarkers on a common timeline, which can, in turn, provide an extra dimension for identifying informative topological descriptors over and above using late-stage information alone.

In this paper, we set out to ameliorate the aforementioned limitations in network-based models of neurodegenerative disease by revealing the combinations of topological descriptors that best explain the temporal evolution of pathology. We introduce a new method that uses the Gaussian Process (GP) Progression Model of *Lorenzi et al. (2019)* to assess candidate descriptors against the full time course of the disease, rather than just late-stage pathology. Moreover, the method identifies a characteristic combination for each disease that defines a novel disease-specific *topological profile*. We extend this concept further to find personalized topological profiles for each individual and demonstrate the consistency of individual profiles with the corresponding cohort profile as well as characterize within-disease variability of individual topological profiles. We focus on patterns and trajectories of atrophy accumulation informed by structural MRI and use three distinct data sets across the spectrum of neurodegeneration (Alzheimer's disease – AD, primary progressive multiple sclerosis – PPMS, and broadly-healthy ageing represented by data collected from community-dwelling ageing individuals – HA). We show that: a) a combination of topological descriptors consistently explains the data better than the best single descriptor; b) the profiles differ substantially among conditions; c) individual profiles cluster around corresponding cohort-level profiles, but only when the profile is estimated from the full time course rather than end-stage only; and d) positioning of individual profiles with respect to each cohort profile is associated with relevant clinical features, thereby potentially benefiting early diagnosis and stratification.

## Results

*Figure 1* gives an overview of the methods used to compute topological profiles at both cohort and individual levels: full details are provided in the Materials and methods section. In this section, we show first that the GP Progression Model estimates trajectories of atrophy evolution that reflect observed atrophy patterns in each cohort (AD, PPMS and HA). Secondly, we show that the topological profiles derived from such atrophy evolution patterns are distinct for each condition and explain observations better than profiles estimated when using end-stage data only, and better than single topological descriptors. Further, we demonstrate that most individual profiles reflect their corresponding cohort profile (and neurological condition), and that those that do not ('outliers') show clinical characteristics of the topologically-nearest cohort.

### Temporal patterns estimated by the GP progression model confirm observed atrophy progression patterns

*Figure 2*, top row, shows the spatio-temporal evolution of the atrophy patterns in each cohort (AD, PPMS and HA) estimated by the GP Progression Model over 41 bilateral regions of interest, obtained by symmetrizing the 82 anatomical regions from the segmentation procedure in *Desikan et al. (2006)* (see Materials and methods: Data description). Each panel shows four temporal stages sampled at uniform intervals according to the estimated disease time of the GP Progression Model (see Materials and methods: GP Progression Model). Each topological profile specifies the rate of pathology accumulation in each area. So, while the rates are time-independent, they vary spatially so that the pattern itself is time dependent. *Figure 2—figure supplement 1* shows a higher temporal resolution visualization of the same progressions. The regional trajectories and the individual time parameters estimated by the model are shown in *Figure 2—figure supplement 2*, *Figure 2—figure supplement 3* and *Figure 2—figure supplement 4*.

In the AD cohort, the first regions to show atrophy are the superior temporal region and the hippocampus, followed by the amygdala, the remaining temporal regions, the insular and the supramarginal regions, and then the precentral and postcentral regions and the posterior lobe. The estimated progression gradually involves the occipital lobe, the middle frontal region, and finally the remaining subcortical areas, with thalamus and caudate last, which matches well-known atrophy progression patterns observed in AD from post-mortem histology (*Eshaghi et al., 2018b*) and in-vivo disease progression models (*Donohue et al., 2014*; *Fonteijn et al., 2012*; *Hinrichs et al., 2011*; *Desikan et al., 2006*). In PPMS, the progression first involves some subcortical areas (caudate, thalamus, pallidum), followed by the superior parietal region, the remaining subcortical areas (amygdala,

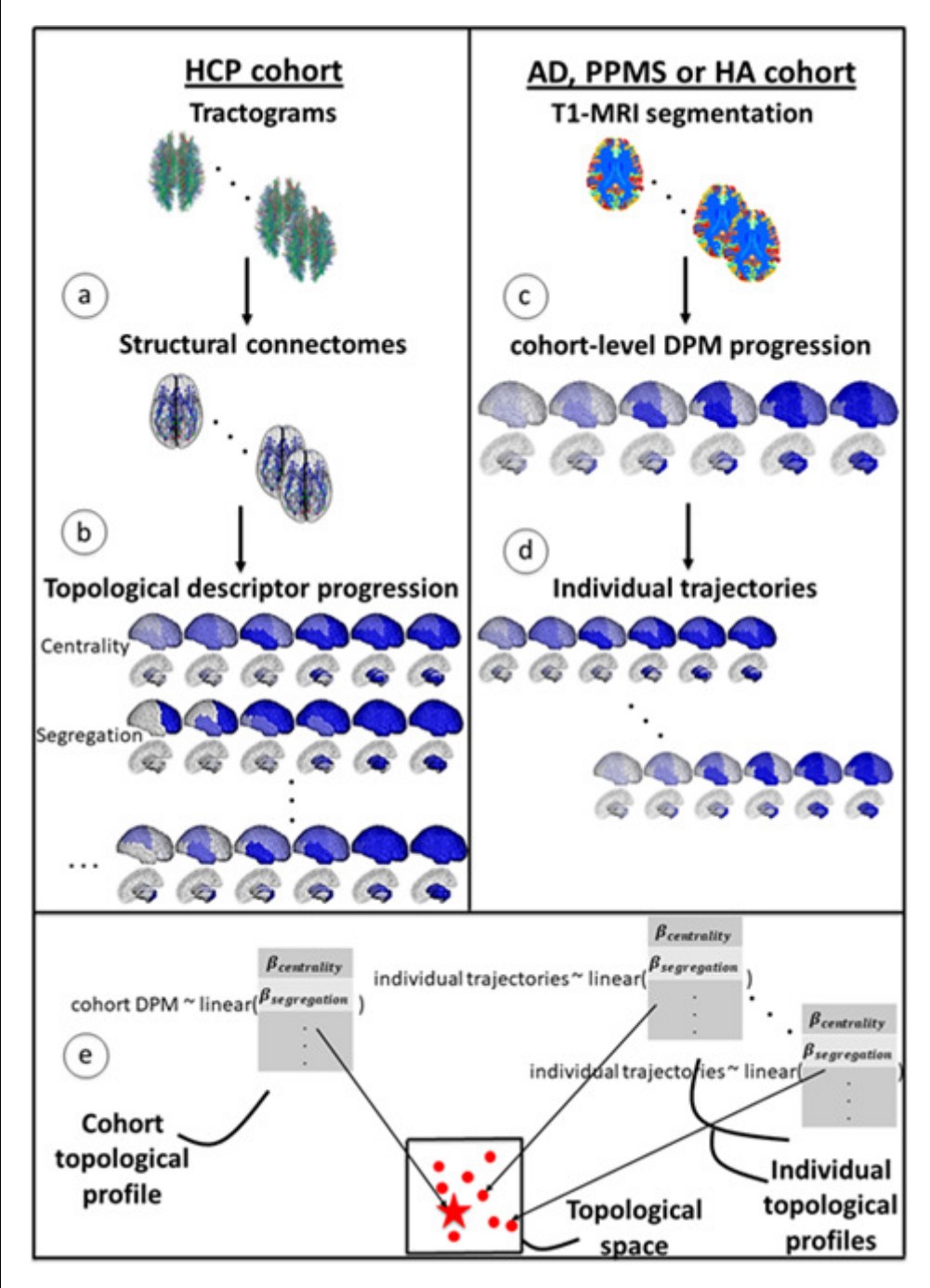

**Figure 1.** Overview of topological profile estimation. (**a**) We construct the average structural connectome from Human Connectome Project (HCP) tractograms; (**b**) we compute topological descriptors on the structural connectome and the progression pattern that corresponds to each; (**c**) we estimate the long-term atrophy progression pattern and its variability within each condition, using GP Disease Progression Model on regional volumes from T1-weighted MRI; (**d**) we estimate rates of progression for each individual from the cohort-level GP progression model; (**e**) we estimate each topological profile (both cohort-level and individual) as the linear combination of topological descriptors, with weights $\beta$, that best matches the observed progression rates. Those profiles are then visualized in a low-dimensional projection of the space of topological descriptors.

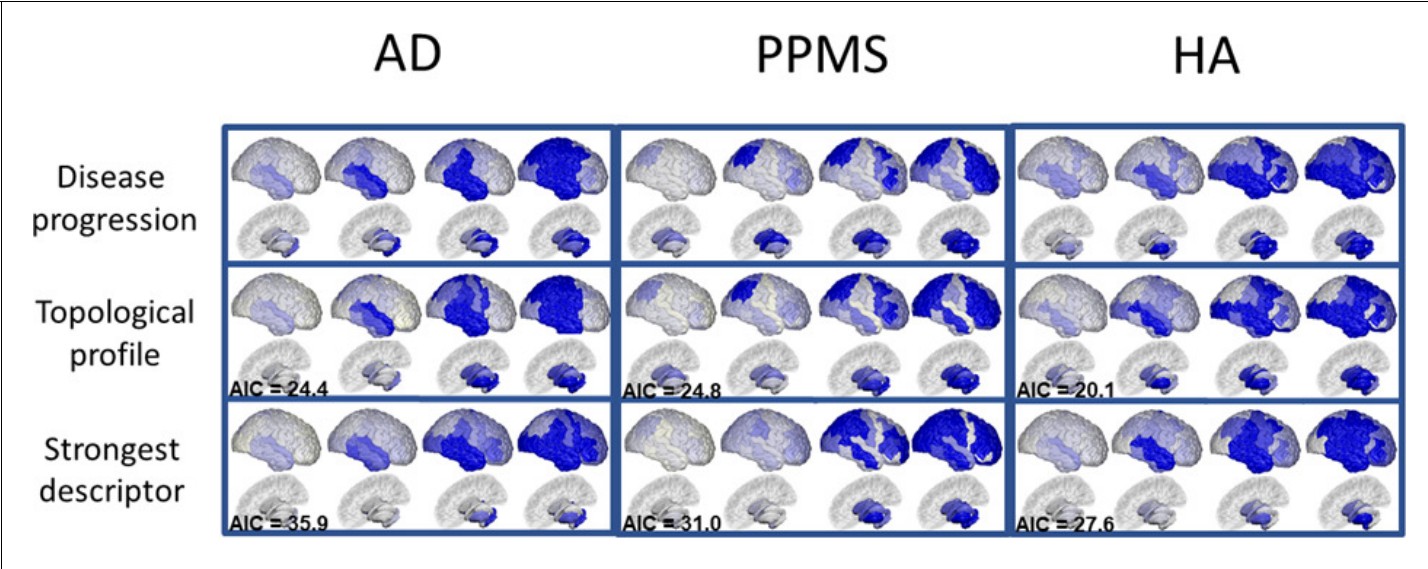

**Figure 2.** Temporal evolution of brain loss in AD, PPMS and HA confirm known atrophy progression patterns, and the progression patterns corresponding to the topological profiles for the three cohorts match the progression of atrophy better than the single best fitting topological descriptor. Top row: 4D representation of the GP disease progression model for AD (left), PPMS (middle) and HA (right). Second row: 4D representation of the progression pattern corresponding to the topological profile for AD (left) - a combination of centrality measures and network-based proximity; PPMS (middle) – a combination of centrality, segregation and cortical proximity measures; and HA (right) - a combination of centrality, cortical proximity, and constant progression. Third row: 4D representation of the progression pattern corresponding the single best fitting topological descriptor for AD (left) - network proximity; PPMS (middle) - segregation; and HA (right) - cortical proximity. Each region's color opacity is proportional to the cumulative abnormality of each region (strong blue means strongly atrophied), and time increases from left to right. AIC is the Aikake Information Criterion for the fit to the observed disease progression (lower is better).

The online version of this article includes the following figure supplement(s) for figure 2:

**Figure supplement 1.** Fine-grained representation of *Figure 2*, with 12 stages.
**Figure supplement 2.** Biomarker trajectories, with standard deviations and measurements for the AD cohort.
**Figure supplement 3.** Individual trajectories, with standard deviations and measurements, for the PPMS cohort.
**Figure supplement 4.** Individual trajectories, with standard deviations and measurements, for the HA cohort.

putamen, nucleus accumbens), few frontal regions (mostly middle- and orbito-frontal), the precentral, and then the occipital and temporal lobes, which become abnormal later in the progression. The dynamics of progression agree with recent results (*Eshaghi et al., 2018a*; *Vinke et al., 2018*), obtained using a different DPM. In the HA cohort, we observe early involvement of the insula, the superior and middle temporal lobes, the middle frontal, and the putamen. Subsequently, the amygdala, hippocampus and nucleus accumbens are affected, the inferior temporal and more frontal regions, followed by the parietal lobe and the cingulate. This agrees with other studies of volume loss in normal aging (*Narvacan et al., 2017*; *Watson et al., 2016*; *Akaike, 1974*).

## Distinct topological profiles for each neurological condition

The second row of *Figure 2* shows the progression corresponding to the topological profiles that best match the GP Progression Model, for each cohort. The topological profiles are sparse linear combinations of nine network metrics, each representing one of five topological descriptors. They are the four described in *Zhou et al. (2012)* – centrality, segregation, network proximity and cortical proximity – and the constant progression descriptor, quantifying the extent to which the rate of atrophy remains constant throughout the progression (see Materials and methods: Network metrics). *Table 1* shows the weight of each metric in the profile for each condition that best explains the corresponding GP disease progression in *Figure 2* top row. Weights with value below 0.10 are set to zero in the final sparse linear combination defining the topological profiles; those weights are shown in bold in *Table 1*.

*Table 1* shows that the topological profiles for AD, PPMS and HA differ substantially. Specifically, AD shows a concurrency of centrality metrics (40%) and network proximity (60%); PPMS shows a

**Table 1.** Weights of the topological profiles of the three cohorts.

The table reports the weights for each network metric, grouped per topological descriptor. Credible intervals for the weights are given in parentheses. Bootstrapping variation is shown in square brackets. Bonferroni-corrected p-values and effect size for the permutation testing of the null hypothesis are shown in braces. In bold, the weights that have been used to compute the topological profiles.

| Topological descriptor | Network metrics | AD | PPMS | HA |
|---|---|---|---|---|
| | Betweenness centrality | **0.21** (0.17) [0.22 (0.18)] {0.01 (2.85)} | **0.10** (0.08) [0.11 (0.08)] {0.01 (2.72)} | 0.09 (0.06) [0.07 (0.05)] {0.01 (3.16)} |
| Centrality | Closeness centrality | 0.01 (0.02) [0.04 (0.04)] {0.01 (2.90)} | **0.11** (0.11) [0.12 (0.12)] {0.01 (2.34)} | 0.03 (0.05) [0.04 (0.04)] {0.01 (1.79)} |
| | Weighted degree | 0.03 (0.03) [0.02 (0.02)] {0.01 (3.10)} | 0.07 (0.05) [0.07 (0.05)] {0.01 (2.96)} | **0.13** (0.09) [0.11 (0.07)] {0.01 (2.78)} |
| | Clustering coefficient | **0.19** (0.11) [0.21 (0.12)] {0.01 (3.32)} | **0.14** (0.05) [0.14 (0.06)] {0.01 (4.52)} | **0.10** (0.07) [0.08 (0.06)] {0.01 (2.97)} |
| Segregation | Inverse degree | 0.05 (0.06) [0.05 (0.05)] {0.05 (0.26)} | **0.17** (0.12) [0.16 (0.11)] {0.01 (3.43)} | 0.01 (0.01) [0.01 (0.01)] {0.01 (5.31)} |
| | Inverse clustering | 0.01 (0.03) [0.05 (0.04)] {0.01 (1.90)} | <u>0.32</u> (0.22) [0.36 (0.24)] {0.01 (2.82)} | 0.01 (0.01) [0.01 (0.02)] {0.01 (2.01)} |
| Network proximity | Shortest path | <u>0.65</u> (0.39) [0.54 (0.35)] {0.01 (3.62)} | 0.06 (0.06) [0.07 (0.06)] {0.01 (2.48)} | 0.01 (0.04) [0.01 (0.02)] {0.01 (1.48)} |
| Cortical proximity | Spatial distance | 0.06 (0.08) [0.10 (0.06)] {0.01 (2.14)} | **0.22** (0.14) [0.21 (0.18)] {0.01 (3.22)} | <u>0.64</u> (0.38) [0.54 (0.32)] {0.01 (3.46)} |
| Constant progression | Constant term | 0.07 (0.02) [0.12 (0.03)] {0.01 (4.01)} | 0.08 (0.04) [0.10 (0.05)] {0.01 (3.56)} | **0.19** (0.09) [0.18 (0.12)] {0.01 (3.58)} |

more complex profile with presence of centrality (35%), segregation (45%) and cortical proximity (20%); HA matches a combination of centrality (25%), cortical proximity (60%) and constant progression (20%). Credible intervals in the topological profile weights, shown in parentheses in *Table 1*, reflect the credible intervals explicitly estimated by the GP Progression Model. Variability of the topological profile parameters under bootstrapping with 100 samples are shown in brackets. The reported p-values and effect sizes (in braces) are relative to the null hypothesis of $\beta = 0$ aside from the term associated with the constant progression, computed via permutation testing and Bonferroni-corrected for multiple comparison across the set of network metrics. All p-values were found <0.01 apart from the inverse degree for AD, for which p=0.048. *Supplementary file 1* - Table S1 shows the topological profiles for all 82 brain regions; results remain consistent. *Supplementary file 1* - Table S2 shows the topological profiles for two subsets of the HA cohort, age-matched with the AD and PPMS cohorts, as compared to topological profiles for the whole HA cohort; results remain consistent.

## Topological profiles match disease progression better than any single descriptor

The third row of *Figure 2* shows predicted atrophy progression using the best-matching single topological descriptor. Overall, the combinations of descriptors (*Figure 2*, second row) match the data (*Figure 2*, first row) more closely than the single best-fitting descriptor (*Figure 2*, third row). For example, according to the topological profile prediction, the parietal lobe is involved in the early stages of AD, in agreement with the data, while it appears to be involved at a later stage, according to the single best-matching descriptor (network proximity, underlined in *Table 1*), which also

underestimates the involvement of the subcortical areas. Similarly in PPMS, the topological profile prediction reproduces the subcortical involvement better than the best-fitting single descriptor (segregation – inverse clustering). In HA, the topological profile prediction shows involvement of the temporal and frontal lobes, as does the atrophy pattern, while the strongest single descriptor (cortical proximity) underestimates subcortical involvement and overestimates parietal involvement. Further, we note that topological profiles explain variance in the data better than the best-fitting single descriptors. Indeed, in the AD cohort, the topological profile explains 82% of the variance, with the constant term explaining just 6%, in contrast to 51% explained by network proximity. In the PPMS cohort, the topological profile explains 83%, the constant term 7%, and inverse clustering 25%. Similarly, in the HA cohort the topological profile explains 88%, the constant term 16%, and cortical proximity 64%. To quantify and compare how well the topological model predictions match the data, we also calculate the Akaike Information Criterion (AIC), which penalizes model complexity (*Maaten and Hinton, 2008*). The topological profiles always provide lower AIC scores than the best-fitting single descriptor model (see *Supplementary file 1* - Table S4), thus explaining the data better without overfitting. For full details please refer to the Materials and methods section.

## Individual profiles group around cohort profiles and separate in topological space

*Figure 3* plots the cohort-level topological profiles and the individual topological profiles on different topological spaces, for different groups of individuals. Personalized topological profiles come from fitting the best combination of network metrics to individual progression rates. The GP Progression Model provides individual time parameters positioning the individuals along the estimated progression and thus enables the estimation of individual progression rates with respect to the global disease progression (see Materials and methods: Personalized topological profiles). We consider a 5D topological space spanned by each of the five descriptors we consider (centrality, segregation, network proximity, cortical proximity and constant progression). The position of a particular topological profile within this space has coordinates that are the sums of the weights of the network metrics corresponding to each descriptor. The ternary plots in *Figure 3(a), (c) and (e)* show the distribution of individual profiles colored according to the cohort to which the individual belongs (red for AD, green for PPMS, blue for HA). The position corresponds to the relative Euclidean distances of the individual profile from each cohort-level profile in the 5D topological space. Thus, the corners correspond to exact matches with one of the cohort profiles, while points at the center of the triangle are equidistant to all three cohort-level profiles. Individual profiles that are closer to the profile of another cohort are highlighted as black diamonds with border color reflecting the true cohort; we refer to these individuals as 'outliers'. *Figure 3(b), (d) and (f)* show projections of the 5D topological space to 2D using t-Distributed Stochastic Neighbor Embedding (tSNE: *Zhou et al., 2012*), which visualizes high dimensional points in a low dimensional space in a way that retains pairwise similarity with high probability, that is most points that are close/distant in the 5D space are close/distant in the 2D visualization. Thus, while the global shapes of the distributions of points have limited interpretation, disconnected groups of points in the tSNE plot reflect separation in the native space. Each tSNE plot shows the cohort-level topological profile from each full cohort (big stars), the variation of the cohort-level topological profile under bootstrap resampling (small stars), and each individual topological profile (dots).

Figure 3(a) and (b) show the plots for topological profiles computed using the GP Progression Model. They include points for only disease-diagnosed individuals in the disease cohorts (AD or MCI in the AD cohort; PPMS-diagnosed in the PPMS cohort; excluding controls in both), but all individuals from the HA cohort. *Figure 3(c) and (d)* show only the controls from the AD and PPMS cohort, as well as all the HA cohort, also using the topological profiles obtained from the full temporal trajectories. *Figure 3(e) and (f)* show the same individuals as *Figure 3(a) and (b)* but using topological profiles derived from only late-stage patient data, mimicking current standard models, which ignore disease progression; see *Supplementary file 1*: Late-stage atrophy modeling. *Supplementary file 1*- Table S3 shows results using random networks to show that the separation is genuinely driven by the underlying structural connectivity.

Figure 3(a) and (b) show that (i) the cohort topological profiles are consistent under bootstrap cross-validation, (ii) the three cohorts separate well in topological space, and iii) individual profiles group around cohort profiles.

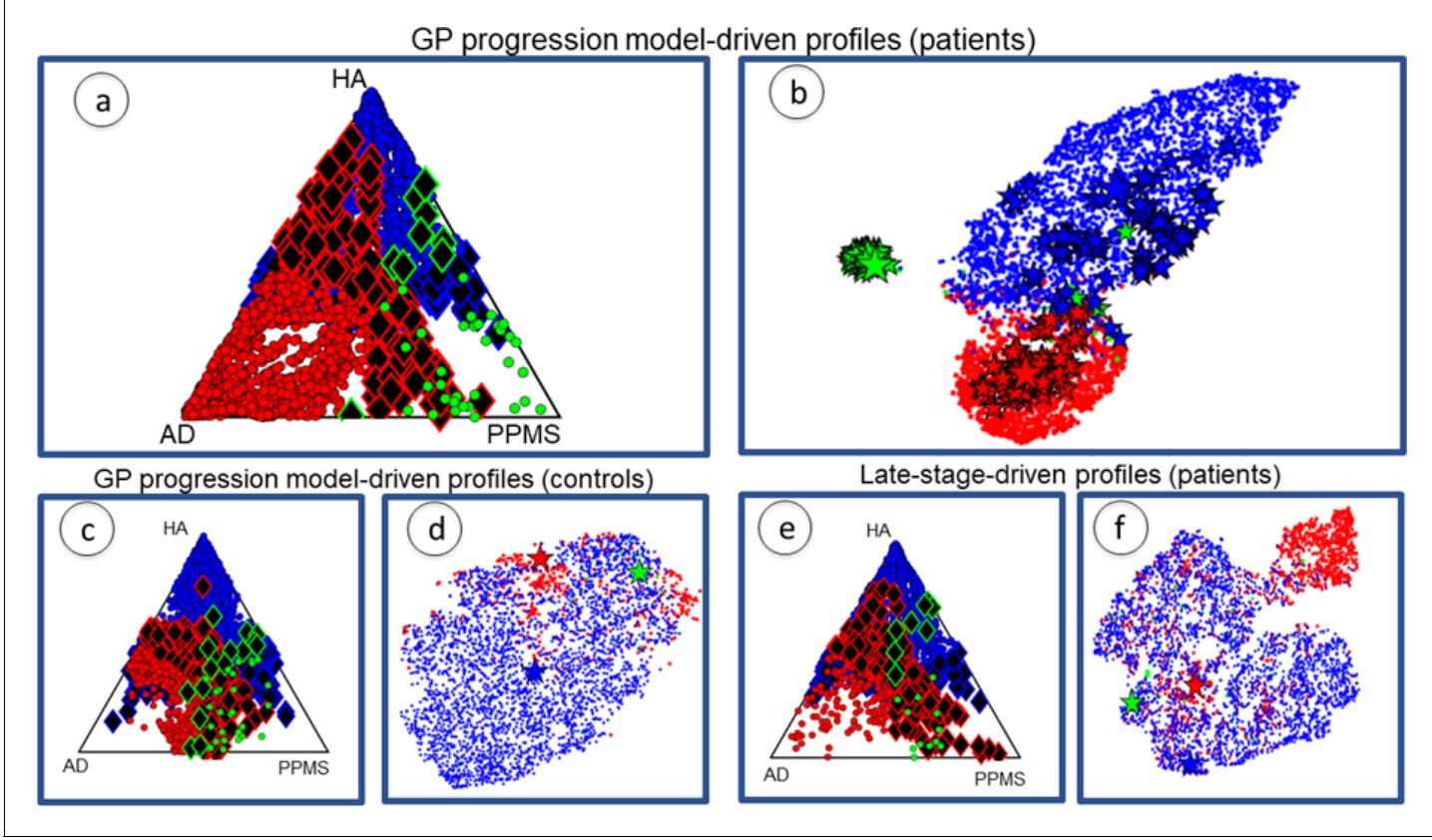

**Figure 3.** Individual profiles are specific for each neurological condition. Red indicates AD individuals, blue indicates HA and green is PPMS. Panel (a) shows ternary plot of the individual profiles, obtained via the GP Progression Model, for AD+MCI-diagnosed individuals, PPMS-diagnosed individuals, and HA individuals plotted according to the distance from the cohort-level profile; corners are cohort-level profiles. Outliers of each cohort are highlighted (identified with diamonds); (b) shows a 2D representation of the topological profiles in (a), using tSNE; big stars represent the cohort profiles and small ones the bootstrapped cohort-profiles; (c) ternary plot using GP Progression Model -driven profiles for only healthy control individuals in the AD and PPMS cohorts, and HA individuals; (d) tSNE plot of data in (c); (e) ternary plot of the individual profiles for AD+MCI-diagnosed, PPMS-diagnosed, and HA individuals, estimated from only late-stage data; (f) tSNE plot for the topological profiles of data in (e). The online version of this article includes the following figure supplement(s) for figure 3:

**Figure supplement 1.** Longitudinal information for the study cohort (AD: 1713 individuals, HA 5463 individuals, PPMS 64 individuals).

*Figure 3(c) and (d)* and *Supplementary file 1*- Table S4 confirm that the separation observed in *Figure 3(a) and (b)* is due to disease-related information, and not to differences in MRI centres, MRI scanner or acquisition protocol, as the three groups of healthy controls show weaker separation than the distinct disease groups.

## Matching progression pattern defines profile better than late-stage atrophy alone

*Figure 3(e) and (f)* show that when using only late-stage atrophy to identify topological profiles: i) the cohorts separate less strongly than using the DPM topological profiles (see *Supplementary file 1*- Table S2); and ii) the rate of assignment of AD and PPMS individual profiles to the corresponding cohort profile is lower than using GP Progression Model-driven topological profiles and with a greater fraction of outliers (see Table 3). Both observations suggest that late-stage information alone is not sufficient to provide distinct topological profiles.

*Table 2* shows the weights for the topological profiles when using late-stage data. We observe that, with respect to the topological profiles estimated using the GP Progression Model (*Table 1*), the centrality components in AD and HA are decreased (from 40% to 25% and from 25% to 15%, respectively), including an increase in uncertainty for all the estimates. In HA, we also note an increase in the constant propagation term (from 15% to 30%), which may be explained as a

**Table 2.** Weights of the topological profiles of the three cohorts when using only late-stage atrophy data are more uncertain and overlap more.

The table reports the weights for each network metric, grouped per topological descriptor.

| Topological descriptors | Network metrics | AD | PPMS | HA |
|---|---|---|---|---|
| | Betweenness centrality | 0.15 (0.11) | 0.11 (0.09) | 0.05 (0.02) |
| Centrality | Closeness centrality | 0.01 (0.03) | 0.06 (0.07) | 0.01 (0.02) |
| | Weighted degree | 0.03 (0.04) | 0.11 (0.08) | 0.09 (0.05) |
| | Clustering coefficient | 0.07 (0.10) | 0.11 (0.15) | 0.03 (0.02) |
| Segregation | Inverse degree | 0.06 (0.06) | 0.12 (0.09) | 0.06 (0.06) |
| | Inverse clustering | 0.09 (0.08) | 0.25 (0.22) | 0.01 (0.04) |
| Network proximity | Shortest path | 0.30 (0.22) | 0.05 (0.05) | 0.02 (0.04) |
| Cortical proximity | Spatial distance | 0.12 (0.09) | 0.15 (0.09) | 0.15 (0.09) |
| Constant progression | Constant term | 0.04 (0.05) | 0.10 (0.07) | 0.33 (0.21) |

compensatory effect for the decrease of the other weights, as the constant propagation term is analogous to the intercept in linear regression – for details see Materials and methods section.

*Table 3* is a confusion matrix of classification rates for subject assignments: without parentheses are the rates for assignment using the GP Progression Model (using patient data only — no controls); numbers in parentheses are the rates when using late-stage information (patients only). These confirm that superior classification rates come from using GP Progression Model driven profiles.

## Alzheimer's disease individuals that exhibit individual profiles closer to the healthy aging topological profile perform better in cognitive tests

*Figure 3(a)* identifies 159 AD-cohort outliers (130 MCI, 29 probable AD) closer to the HA cohort-level profile, out of 1312 MCI+AD patients. We analyzed clinical and demographic information for the AD outliers with respect to the rest of the AD cohort (see *Supplementary file 1* Table S5a). An ANOVA 1-way test between the two groups was performed and differences were found in MMSE, where the outliers exhibit a higher MMSE score ($27.6 \pm 1.2$ vs $26.1 \pm 1.7$). Slight differences were found in the Clinical Dementia Rating Scale, with outliers scoring lower ($1.8 \pm 0.7$ vs $2.1 \pm 0.3$). Both results show that the outliers have reduced cognitive deficits compared to inliers, on average. Further information on the ANOVA results is provided in the *Supplementary file 1* Table S5a. The same analysis performed on the outliers defined by late-stage-driven topological profiles did not report any significant differences between the groups, suggesting that outliers defined using the GP Progression Model-driven topological profiles depart more genuinely from the disease cohort and phenotype of typical amnestic AD; see the *Supplementary file 1* Table S6a).

## Healthy ageing individuals that exhibit individual profiles closer to the AD topological profile show signs of prodromal dementia

*Figure 3(a)* identifies 358 of 5463 HA individuals closer to the AD cohort profile. An ANOVA 1-way test between these outliers and the rest of the HA group found differences in MMSE, where the outliers exhibit a lower MMSE score ($27.7 \pm 0.8$ vs $28.0 \pm 0.6$); and a difference in age, with the outliers being older ($70.5 \pm 10.1$ vs $64.5 \pm 9.8$). No significant difference in gender or APOE status (see *Supplementary file 1* Table S5c) was found. In order to investigate whether the increased cognitive deficit in the HA outliers indicates an actual prodromal phase of dementia, or is just an age-related effect, we analyzed the incidence, in the outlier group, of the individuals that were healthy at baseline, but developed dementia after 2–4 years (see Materials and methods: Participants - HA). Of those 148 individuals, 105 reside in the HA-AD outlier group, which thus contains 70% of the individuals with prodromal dementia; in contrast the non-outlier group consists of only 5% prodromal dementia cases. The same analysis performed on the outliers defined by late-stage-driven topological profiles reports only a group difference in age with older outliers ($65.2 \pm 10.9$ vs $60.0 \pm 7.0$). No other significant differences were found (see *Supplementary file 1* Table S6c).

**Table 3.** Confusion matrix of classification rates for individuals' assignment to each cohort by matching individual topological profiles to cohort-level topological profiles.
Without parentheses: results using full GP Progression Model -driven topological profiles. Within parentheses: results using topological profiles estimated from only late-stage data. Higher numbers are better in diagonal entries (correct assignment), and lower numbers are better in off-diagonal entries (incorrect assignment).

|      | AD         | PPMS       | HA         |
|------|------------|------------|------------|
| AD   | 84% (57%)  | 4% (9%)    | 12% (34%)  |
| PPMS | 2% (16%)   | 68% (45%)  | 29% (38%)  |
| HA   | 7% (14%)   | 2% (12%)   | 91% (74%)  |

## Primary progressive multiple sclerosis outliers show no significant differences with the rest of the group

We also analyzed PPMS with respect to both AD and HA: the analysis returned 53 AD individuals closer to PPMS; 1 PPMS individual closer to AD; 13 PPMS individuals closer to HA and 100 HA individuals closer to PPMS. No significant differences were found between any subgroups, for any comparisons: we analyzed age, gender, expanded disability status scale (EDSS) and disease duration when looking at PPMS outliers in AD or HA; age, gender, APOE4 and MMSE score when looking at HA outliers in PPMS; age, gender, years of education, MMSE, ADAS-Cog, APOE4, CDRSB, AV45, when looking at AD outliers in PPMS. Details can be found in *Supplementary file 1* Table S5b). The same analysis on late-stage-driven topological profiles also returned no significant differences (*Supplementary file 1* Table S6b).

## Features derived from the topological profile correlate with clinical features

Finally and more generally, we analyzed features extracted from topological profiles, in particular, the distance of individual profiles from the cohort profile, and show that they correlate negatively with individual cohort-specific clinical features. Specifically, MMSE for the AD cohort (R = 0.11, p<0.01); EDSS for PPMS (R = 0.68, p=0.07); and age for HA (R = 0.32, p<0.01).

## Discussion

We have introduced a novel method to estimate cohort and individual-level topological profiles of neurodegeneration using computational disease progression models in combination with imaging data sets. The profiles give new insight into the relationship between brain connectivity and the progression pattern of neurodegeneration. We demonstrate the ideas using three cohorts representing different neurological conditions: AD, PPMS, HA. We showed that combinations of topological descriptors explain observations better than any individual descriptor and that the combinations representative of each condition, despite some commonality as suggested in *Soto and Pritzkow (2018)*, *Jucker and Walker (2018)* and *Fu et al. (2018)*, differ substantially both at the cohort and individual level. We emphasize that these differences go beyond simple observations that the pattern of atrophy accumulation is different in these three cohorts, which could arise simply from equivalent spreading mechanisms from different epicentres; our results go further by strongly suggesting distinct modes of dependence of the pathological spread on the underlying connectivity. We also show that using the full disease time-course, estimated via a GP Progression Model (*Desikan et al., 2006*) produces better-defined topological profiles than using only late-stage atrophy. This highlights a key weakness in previous state-of-the-art results using late-stage atrophy alone, which can be ameliorated using recent advances in disease progression modeling. Further, we retrieved significant correlation between features of the topological profiles and individual clinical or demographic features, suggesting potential clinical utility of the topological profile. Finally, we showed that the outliers in the AD-HA comparison display characteristics that align them with the other cohort, providing evidence that our topological profiles reflect underlying disease processes, and indicating potential use of the topological profile to highlight disease risk.

As discussed in the introduction, one key implication here (as in previous literature *Raj et al., 2012*; *Raj et al., 2015*; *Seeley et al., 2009*; *de Haan et al., 2012*) is that graph-theoretic measures may associate with different mechanisms of disease spread in neural networks. Specifically, centrality represents nodal stress (vulnerability of brain hubs), segregation trophic failure (vulnerability of isolated regions), network proximity transneuronal spread of a prion-like agent along network connections, and cortical proximity unguided diffusive propagation (*Zhou et al., 2012*). While we draw on these potential links in the following sections, it is important to note that the links between the mathematical abstraction and the biological mechanisms are simplistic, and graph-theory metrics do not fully capture or explain the range of potential spreading mechanisms.

## The centrality component in AD topological profiles becomes apparent when taking into account the temporal evolution of atrophy

The AD topological profile identified here supports recent results that identify neuronal distance from epicentre (or trans-neuronal spread) as the principal topological descriptor of neurodegeneration in AD (*Jucker and Walker, 2018*; *Jucker and Walker, 2013*; *Cope et al., 2018*; *Raj et al., 2015*; *Weickenmeier et al., 2019*). However, both cohort- and individual-level profiles also show a substantial additional component of neurodegeneration due to centrality metrics — vulnerability of brain network hubs. This component has not clearly been highlighted previously, because a) it is less apparent in a topological profile estimated from only late-stage atrophy data, and b) it is suppressed when using the previous single-descriptor approach. We found contrasting importance of centrality metrics between full-disease-course and end-stage-only profiles. This might arise from metabolic demand increasing primarily during early degeneration so that hub-vulnerability drives early phases of the progression pattern (*Alladi et al., 2007*), while neuronal distance from the epicentre dominates later stages. This finding might also explain some aspects of the biological heterogeneity of AD, which produces high variance in atrophy patterns across individuals. In particular, the early centrality component of the topological profile suggests that pathology may start around local hubs, which is consistent with the focal presentation of atrophy in atypical Alzheimer's diseases such as posterior cortical atrophy or progressive aphasia (*Du, 2006*; *Mendez et al., 2002*; *Barnham et al., 2004*). Once established, each disease then spreads through the same broad networks, leading to syndromic convergence in advanced stages of AD.

Similarly, in HA and PPMS, hub vulnerability appears as a significant component only when considering the full trajectory rather than only late-stage atrophy. This may suggest that activity-dependent mechanisms, such as oxidative stress (*Cagnin et al., 2001*), microglial activation (*Hickman et al., 2018*; *Chiaravalloti and DeLuca, 2008*), neurovascular dysfunction (*Sweeney et al., 2018*) or 'virtual hypoxia' (*Trapp and Stys, 2009*) influence early brain loss across a range of neurodegenerative conditions.

More generally, the differences we observe between full-disease-course and end-stage-only profiles emphasize the importance of considering the full temporal trajectory in assessing the likelihood of potential propagation mechanisms.

## Degree of alignment of individual profiles with cohort profiles informs on patient status

Our results indicate that departures of individual profiles from that of the corresponding cohort may signal abnormality with respect to their cohort, while alignment with another cohort-level profile may correspond to clinical features of that other cohort. For example, the AD individuals with a topological profile close to the HA cohort-level profile show on average lower cognitive deficit than the full AD cohort. On the other hand, HA individuals with topological profiles that tend towards the AD cohort-level profile exhibit reduced MMSE score, suggesting cognitive impairment; also, a substantial proportion (70%) of the HA prodromal-dementia individuals (who developed dementia in 2–4 year follow-up) lie in the outlier group, suggesting evidence of neurodegenerative mechanisms consistent with AD. We emphasize again that we identify these outliers using the topological descriptors that explain the atrophy pattern, and not on the atrophy pattern itself, with the intention of highlighting individuals showing abnormal mechanisms of pathology propagation with respect to their cohort.

We found no significant difference in clinical features between PPMS outliers, tending towards either AD or HA, compared with the rest of the PPMS group: topological profiles seem to identify differences among PPMS subjects that are invisible to the available demographic and clinical features (age, gender, EDSS and diseases duration). However, sample sizes are low (N = 1 PPMS-AD outlier; N = 13 PPMS-HA outliers). Further investigation on a larger cohort of progressive MS subjects, as well as with a broader class of clinical features, may clarify whether the variation we observe in topological profiles arise from spurious effects, or are genuinely informative on the clinical status of the outliers beyond what the available clinical features can identify. For example, future studies of older and longer-term PPMS patients could test the hypothesis that individual topological profiles may inform on co-morbidities, for example for an MS-patient who shows signs of dementia, their profile position with respect to the AD cohort profile may inform on whether the dementia comes from MS-related neurodegeneration or AD comorbidity. However, we cannot test this hypothesis here, as our PPMS cohort are 'early' PPMS (they were recruited to this study within 5 years of diagnosis), and no subject reported dementia as would be expected (*Maier-Hein et al., 2017*), although a detailed cognitive assessment was not performed, this agrees.

## Neurodegeneration in AD and PPMS is not simply an acceleration of the aging process

Overall, our results suggest that neurodegeneration in AD and PPMS is not simply an acceleration of the aging process, but that the two diseases have distinct topological profiles of neurodegeneration, which differ from those of HA. The cohort and individual-level topological profiles we identify for AD and PPMS are clearly distinct from one another. The AD topological profile is a combination of centrality and network proximity, while the PPMS profile has diverse contributing mechanisms including centrality, segregation and cortical proximity, suggesting that many different processes are at play. Also, both separate from the HA topological profile, which has a strong component of both the cortical proximity and the constant propagation mechanisms, suggesting relatively uniform loss across the brain that is not linked to any particular brain-connectivity feature.

Further, we note that we have regressed out the effects of ageing in the AD and PPMS cohort in order to minimize the effects of the ageing process on disease models. Nevertheless, some components of ageing might still appear in the model for both, which is one key reason for including the HA model for comparison.

These observations underline the idea that mechanisms, in addition to actual atrophy patterns, are distinct in these diseases from those underpinning normal aging.

## Future work

Here, we provide a proof of concept of the idea of topological profiling through image-analysis and temporal disease progression modeling, and a demonstration of its potential utility. Multiple opportunities arise for future refinements of the methods, as well as extensions and applications of the key ideas.

Methodologically, several steps in the processing pipeline can limit the accuracy of the network metrics in representing neurodegenerative mechanisms. For example, structural connectome estimation using tractography is prone to false positive and negative connections (*Thomas et al., 2014*; *Petersen et al., 2017*) that can influence subsequent predictions. Nevertheless, tractography does reliably recover at least many of the major known anatomical connections sufficiently well to highlight broad differences in atrophy patterns associated to the different mechanisms. Anatomically constrained probabilistic tractography coupled with SIFT-ing should provide more accurate predictions than deterministic tractography (*Thomas et al., 2014*; *Descoteaux et al., 2009*; *Tsai, 2018*; *Daducci et al., 2016*; *Fornari et al., 2019*), but future work might consider better ways to mitigate errors for example by quantifying uncertainty in tractography output.

Here, we take an average connectome over multiple young and healthy individuals as the substrate for pathology propagation. By doing so, we do not account for any influence of the pathology on the connectome itself (*Oxtoby et al., 2018*), as some other models attempt to *Iturria-Medina and Evans (2015)*; *Fornito et al. (2015)*; *Oxtoby et al. (2018)*; *Zeighami et al. (2015)*; *Powell et al. (2018)*. However, we believe this to be a reasonable anatomical reference of a 'pure' underlying substrate of propagation to investigate grey matter loss and decide to avoid deliberately

the additional complexity of white matter disruptions, which are not well understood and would complicate the model substantially. This decision is also supported by recent results in AD showing that the choice of connectome (young and healthy or subject-specific) does not significantly impact the predictive ability of a model for trans-synaptic transmission of toxic proteins (*Acosta et al., 2018*). However, future models could approximate reductions of connectivity arising from white matter damage commensurate with grey-matter atrophy, but this requires a clear picture of the interaction of such processes, for example from disease progression models of regional grey-matter atrophy and white-matter integrity. We believe using the healthy connectome as the substrate for propagation is a sensible first-order approximation.

One major advantage of our approach for evaluating topological descriptors is that it considers group-level longitudinal information on atrophy rather than considering only late-stage atrophy patterns. Our method uses each network metric to obtain a scalar value at each graph node that provides a template of the rate of pathology accrual at that node compared to others, and then compares to rate-of-change of atrophy evolution as estimated by the GP Progression Model. Future work may extend the idea by accounting for variable change over time, which will need to compare the full temporal evolution of atrophy to the full temporal evolution induced by each topological descriptor. One additional caveat of using the GP Progression Model is that the model captures the trajectory of change in biomarker values rather than underlying pathology; differences among biomarker accuracy (e.g. higher noise in volume measurements of smaller regions) can make the order of biomarker change depart from the order of underlying pathological change, although we expect such effects tend to be small with our processing pipeline – see *Fonteijn et al. (2012)* and *Hinrichs et al. (2011)* for further discussion.

Our approach does not account for heterogeneity within cohorts. In addition to variation in the structural connectome discussed above, the epicentre for disease propagation may vary among patients in a particular disease class and a number of recent works explore methods for per-subject epicentre selection (*Young et al., 2014*; *Young et al., 2018*). Here, we estimated a single cohort-level epicentre as the most atrophied region for each data set, and each reflected common knowledge of brain tissue loss in AD, PPMS and HA. Our epicentres broadly agree with (*Zhou et al., 2012*), who found that individually selected epicentres in a number of neurodegenerative diseases are always seated in or near the most atrophied region. Recent data-driven approaches (*Mezias and Raj, 2017*) further reveal within-cohort heterogeneity of temporal trajectories, which our current approach does not accommodate, but such advances provide great potential for future work to focus topological profiling on more homogeneous subgroups and to explore the variability of topological profiles among different disease subgroups. Indeed, we believe application to broader cohorts, as for instance generic progressive MS including both primary and secondary (which would be more numerous) would require better identification of distinct subgroups than the traditional MS classifications (clinically isolated syndrome, relapsing remitting MS, and primary and secondary progressive MS), and methods such as SuStain (*Mezias and Raj, 2017*) would offer a way to identify within-cohort subtypes for better topological profile estimation. As concerns applications to prodromal phases of diseases – an example being radiologically isolated syndrome for MS – future work will be devoted to build models of the whole neurodegeneration process within specific diseases and their subtypes defined by distinct trajectories of pathological change, which might be used for instance for discriminating subjects that convert to degeneration from others that do not, thereby informing treatment and care choices.

The ideas we propose here extend to a much larger range of diseases and disease subtypes and offer a new and unique tool to exploit the full power of large imaging data sets in exposing mechanisms of disease aetiology and propagation. With further development and experiments, topological profiles may also provide useful information in the clinic. For example, AD-like HA outliers may be considered at risk for AD and decisions could be informed by this, including patient management (more frequent visits to monitor the patient) and recruitment into clinical studies/trials.

The underlying models of propagation remain simplistic and a great many alternative descriptors and mechanisms could easily be included, such as propagation via functional networks (*Zhou et al., 2012*; *Cope et al., 2018*), or different kinds of tractography to represent intra and extra-axonal propagation (*Iturria-Medina et al., 2017*; *Iturria-Medina et al., 2018*; *Oxtoby et al., 2018*). Simplicity of the models and limitations of processing steps is important to bear in mind while interpreting results, and validation, for example against histological analysis in animal models (*Eshaghi et al.,*

*2014*), is an important future target to establish the extent to which mechanistic information inferred in this way matches low-level observations of molecular propagation.

## Materials and methods

Our pipeline for creating topological profiles in a neurodegenerative condition is the following.

1. Generate an average structural connectome from young healthy data.
2. Pre-process the cohort data to adjust for nuisance variables and generate the longitudinal data for regional atrophy and topology, relative to healthy controls.
3. Build the DPM and topological profile.

We now describe the data and these steps in detail.

### Data description

#### Participants – AD

The data used in the preparation of this article were obtained from the Alzheimer's disease Neuro-imaging Initiative (ADNI) database (adni.loni.usc.edu). The ADNI was launched in 2003 as a public-private partnership, led by Principal Investigator Michael W. Weiner, MD. The primary goal of ADNI has been to test whether serial Magnetic Resonance Imaging (MRI), Positron Emission Tomography (PET), other biological markers, and clinical and neuropsychological assessment can be combined to measure the progression of MCI and early AD. For up-to-date information, see www.adni-info.org. We collected longitudinal measurements for all the available ADNI 1/GO/2 individuals with at least one 'quality control' flagged 3D-T1 MRI scan. Repeated T1-weighted structural MRI images were acquired at 3T machines across multiple centers according to a harmonized protocol. Longitudinal FreeSurfer was used to align images from multiple time-points according to subject specific median templates, in order to avoid temporal bias. Subject with no available diagnosis were discarded, thus leaving us 1713 subjects, with 'HC', 'MCI' or 'AD' diagnosis. We collected longitudinal FreeSurfer 5.1 vol data on all the GM and subcortical regions from ADNIMERGE.csv. For information on scanning protocols and segmentation algorithms see www.adni-info.org. The subjects included in our analysis are 1713; the mean (std) age of the cohort is 73.9 (7.2) years. The overall scans are 6670; the average (std) time between scans is 2.4 (1.8) years. More information is in *Figure 3—figure supplement 1* and *Supplementary file 1*- Table S7.

#### Participants – PPMS

This was a retrospective study of 64 participants, studied at the UCL Queen Square Institute of Neurology in London. The participants include 44 individuals with primary progressive multiple sclerosis (PPMS) and 20 healthy controls (HC). We collected longitudinal FreeSurfer 5.1 volumetric data on all the GM and subcortical regions. For information on scanning protocol, segmentation algorithms, please refer to *Hofman et al. (2015)*. The longitudinal data set used for this study consisted in 64 individuals; the mean (std) age of the cohort is 41.6 (10.2) years. The overall scans are 244 scans; the average (std) time between scans is 4.0 (1.5) years. More information are in *Figure 3—figure supplement 1* and *Supplementary file 1*- Table S7.

#### Participants – HA

The data used in the preparation of this article were obtained from The Rotterdam Scan Study (*Ikram et al., 2015*; *Veraart et al., 2016*). The Rotterdam Study is a prospective cohort study ongoing since 1990 in the city of Rotterdam in The Netherlands. The study targets cardiovascular, endocrine, hepatic, neurological, ophthalmic, psychiatric, dermatological, otolaryngology, locomotor, and respiratory diseases. Initially, in 1995 and 1999, random subsamples of participants from the Rotterdam Study underwent neuroimaging, whereas from 2005 onwards brain MRI has been implemented into the core protocol of the Rotterdam Study. We excluded individuals with prevalent dementia at study entry, and individuals presenting cortical infarcts on the MRI. The study contains 148 individuals that would develop dementia in the 2–4 years follow-up(s). We leave them in the study as we are interested to model their individual topological profiles with respect to the topological profile of the cohort-level. We collected longitudinal FreeSurfer 5.1 volumetric data on all the GM and subcortical regions. For information on the scanning protocols and segmentation algorithms see *Veraart et al.*

*(2016)*. The longitudinal data set used for this study consisted in 5463 individuals; the mean (std) age of the cohort is 64.8 (10.8) years. The overall scans are 11627 scans; the average (std) time between scans is 5.3 (1.1) years. More information is in *Figure 3—figure supplement 1* and *Supplementary file 1*- Table S9.

## Data pre-processing to obtain model inputs

On each data set of volumetric GM and subcortical regions separately, we performed three steps:

1. Adjustment for nuisance variables: we constructed a regression model for each region separately with volumes as dependent variable and total intracranial volume, gender, and age (age not included for the HA cohort) as independent variables.
2. Variable selection: from the original FreeSurfer Regions of Interest (ROIs) we discarded white matter, brain stem, ventricular and cerebellar regions, leaving 82 ROIs. Then, we averaged the (adjusted) volumes of each region from both hemispheres, obtaining 41 bilateral regions.
3. Z–scores computation: we computed z–scores against a control population. For the AD data set, the control population was formed by the 'HC'-diagnosed individuals; for the HA cohort it was formed by the 'Young' individuals (those whose age is more than one standard deviation less than the mean age); for the PPMS data set it was formed by the 'HC'-diagnosed individuals.

## HCP participants

Data used in the preparation of this work were obtained from the MGH-USC Human Connectome Project (HCP) database (ida.loni.usc.edu/login.jsp). The HCP project (Principal Investigators: Bruce Rosen, M.D., Ph.D., Martinos Center at Massachusetts General Hospital; Arthur W. Toga, Ph.D., University of California, Los Angeles, Van J. Weeden, MD, Martinos Center at Massachusetts General Hospital) is supported by the National Institute of Dental and Cranio-facial Research (NIDCR), the National Institute of Mental Health (NIMH) and the National Institute of Neurological Disorders and Stroke (NINDS). Collectively, the HCP is the result of efforts of co-investigators from the University of California, Los Angeles, Martinos Center for Biomedical Imaging at Massachusetts General Hospital (MGH), Washington University, and the University of Minnesota. The data set consisted of 24 unique subjects: we collected raw high–resolution 3D T1–weighted and DTI of 24 age and gender–matched subjects (age: 26, 50% female).

## Connectome generation

Structural connectomes were generated using tools provided in the MRtrix3 software package (http://mrtrix.org). The pipeline included (*Oxtoby et al., 2018*): DWI denoising (*Andersson and Sotiropoulos, 2016*), pre-processing (*Tustison et al., 2010*) and bias–field correction (*Modat et al., 2014*); inter–modal registration (*Jeurissen et al., 2014*); T1 tissue segmentation (*Braak and Braak, 1995*); spherical deconvolution (*Smith et al., 2013*; *Tournier et al., 2010*); probabilistic tractography (*Smith et al., 2012*) utilizing anatomically-constrained tractography (*Yeh et al., 2016*), dynamic seeding and SIFT (*Tournier et al., 2010*); T1 parcellation (*Braak and Braak, 1995*); robust structural connectome construction (*Bullmore and Sporns, 2009*). Our anatomical connectome for each participant is a weighted adjacency matrix that includes only inter-node connections across the 82 ROIs consisting of cortical and subcortical gray-matter regions, excluding the cerebellum and brain stem. The average structural adjacency matrix was computed by taking the mean over the subject-wise matrices. Weights, or connection strengths, were normalized to [0, 1]. The inter-hemisphere average is performed after the network metrics computation step (see below).

## Mathematical modeling

### Network metrics

We selected five different descriptors of graph topology (centrality, segregation, network proximity cortical proximity and constant progression) and nine metrics across these categories (*Lehmann et al., 2013*):

1. Centrality measures:
   a. Weighted degree: the sum of the weights of the connections incident the node of interest.

 b. Betweenness centrality: the number of shortest paths between any two nodes that pass through the specific node.

 c. Closeness centrality: the inverse of the path length between the node of interest and all the other nodes.

 d. Clustering coefficient: the proportion of triangular sub-networks formed by the node and its neighboring nodes.

2. Segregation measures:
 a. Inverse of weighted degree.
 b. Inverse of clustering coefficient.

3. Network proximity:
 a. Shortest path to epicenter.

4. Cortical proximity:
 a. Spatial distance from epicenter.

5. Constant progression:
 a. Constant term quantifying to what extent the rate of atrophy remains through-out the progression.

Each network metric produces a scalar value for each node that indicates its vulnerability to pathology relative to all other nodes. Each thus leads to a temporal progression pattern in which the accumulation of pathology, for example the atrophy or rate of change of volume, is proportional to that vulnerability (see *Figure 4*). In order to identify the epicenters needed to compute 3a) and 4a), we selected, for each subject, the region with the highest z-scores of GM volumes (i.e. the 'most atrophied' region) in the last follow-up MRI scan; the cohort-level epicenters are then, for each cohort, the regions that most frequently appear as individual-level epicenters. They are the hippocampus (AD), the caudate (PPMS) and the insula (HA). These epicenters are supported by literature in AD (*Zhou et al., 2012*; *Abe et al., 2008*), PPMS (*Eshaghi et al., 2018a*; *Vinke et al., 2018*) and HA (*Allen et al., 2002*; *Good et al., 2001*; *Kalpouzos et al., 2009*; *Tisserand et al., 2004*; *Rubinov and Sporns, 2010*).

We computed the first seven network metrics using the Brain Connectivity Toolbox (*Daube-Witherspoon and Muehllehner, 1986*) in MATLAB, after appropriate normalization (*weight_conversion* function). To compute the cortical proximity metric, which does depend on the structural connections, we select the segmented structural MRIs in the HCP cohort, and measure the (average) pairwise cortical distances between all the macro-regions' barycenters (using the *mris_pmake* command in FreeSurfer). The constant progression metric, which also does not depend on the structural connections, simply assigns a constant value to each region.

We note that the network metrics computing similar attributes of a graph such as centrality and efficiency, or segregation and inefficiency, are not independent and in fact highly correlated producing strongly overlapping progression patterns. However, we include multiple network metrics for certain topological descriptors to capture variability in the corresponding progression pattern that fits within the definition of the descriptor.

Network metrics were computed on the average structural adjacency matrix formed of 82 ROIs, and then averaged across hemisphere, obtaining an array of 9 metrics for each of the 41 symmetrized regions $v$, $\boldsymbol{g}(v) = (g_1(v), \ldots, g_9(v))^T$, where we set $g_9(v) = 1$ to describe the constant progression term.

## GP progression model

Continuous Disease Progression Models (DPM) aim to estimate the long-term temporal pattern of disease progression directly from short-term longitudinal data sets and to stage patients based on individual observations. The problem is challenging, due to the lack of a well-defined temporal reference in longitudinal data sets: time of onset is usually unknown and rate of pathology accrual is highly variable in most neurodegenerative conditions. DPM techniques typically tackle this problem by assuming that each visit at which measurements are taken occurs at an unknown 'disease time', the particular value of which for each individual subject is a hidden variable estimated while fitting the set of trajectories. The disease-time axis parametrizes the natural evolution of the pathology common to all individuals.

The models usually assume mixed effects, in which the long-term trajectories are the fixed effect (represented by parametric or non-parametric functions), and the individual variations from the

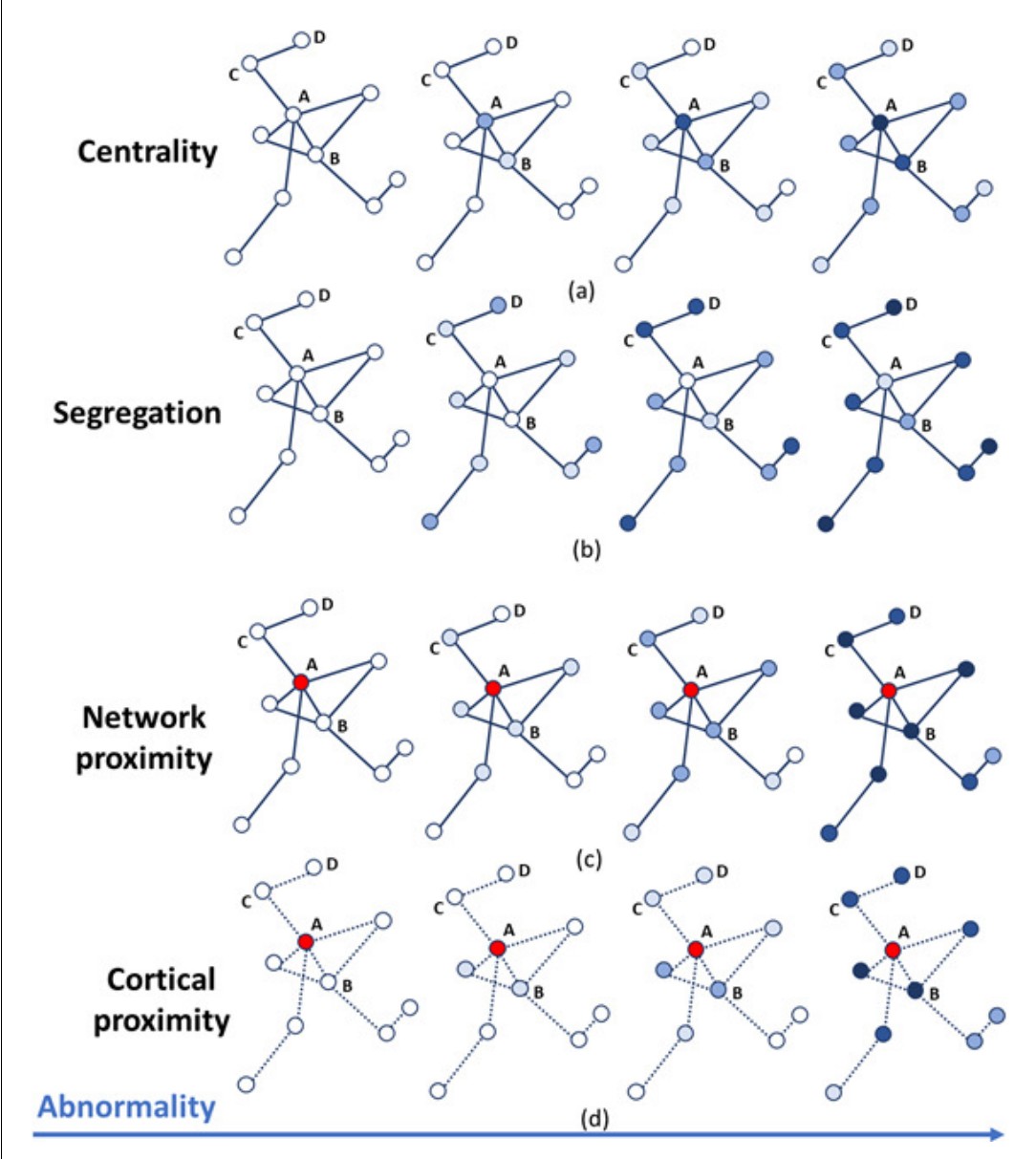

**Figure 4.** Temporal progression patterns (left-to-right) of different descriptors. For each descriptor (row), abnormality increases in a descriptor-specific pattern. The magnitude of cumulative abnormality at a node is proportional to the color intensity. Red nodes are epicenters. (**a**) Centrality: node A is affected first due to having the highest centrality, followed by node B, then C and D. (**b**) Segregation: node D is affected first due to having the highest segregation, followed by C, then B and A. (**c**) Network proximity: nodes B and C are affected before D, because they are closer to the epicenter A (along the connectivity network). (**d**) Cortical proximity: node B is affected first because of its spatial proximity to the epicenter A, then C and finally D. Here edges are dashed as no information is needed from connectivity.

group-level trajectories are the random effects. Further, individual time reparametrization parameters (also known as time warp parameters [*Venkatraghavan et al., 2017*] or disease scores [*Donohue et al., 2014*; *Jedynak et al., 2012*]) are estimated by quantifying the individual positions with respect to the estimated time-frame. Here, we use the GP Progression Model of *Desikan et al. (2006)*, which is a non-parametric Bayesian mixed effect model, to estimate long-term trajectories of regional GM volumes on each data set (AD, PPMS and HA) separately. *Figure 5* shows a pictorial example of the GP Progression Model combined with the topological profiles procedure.

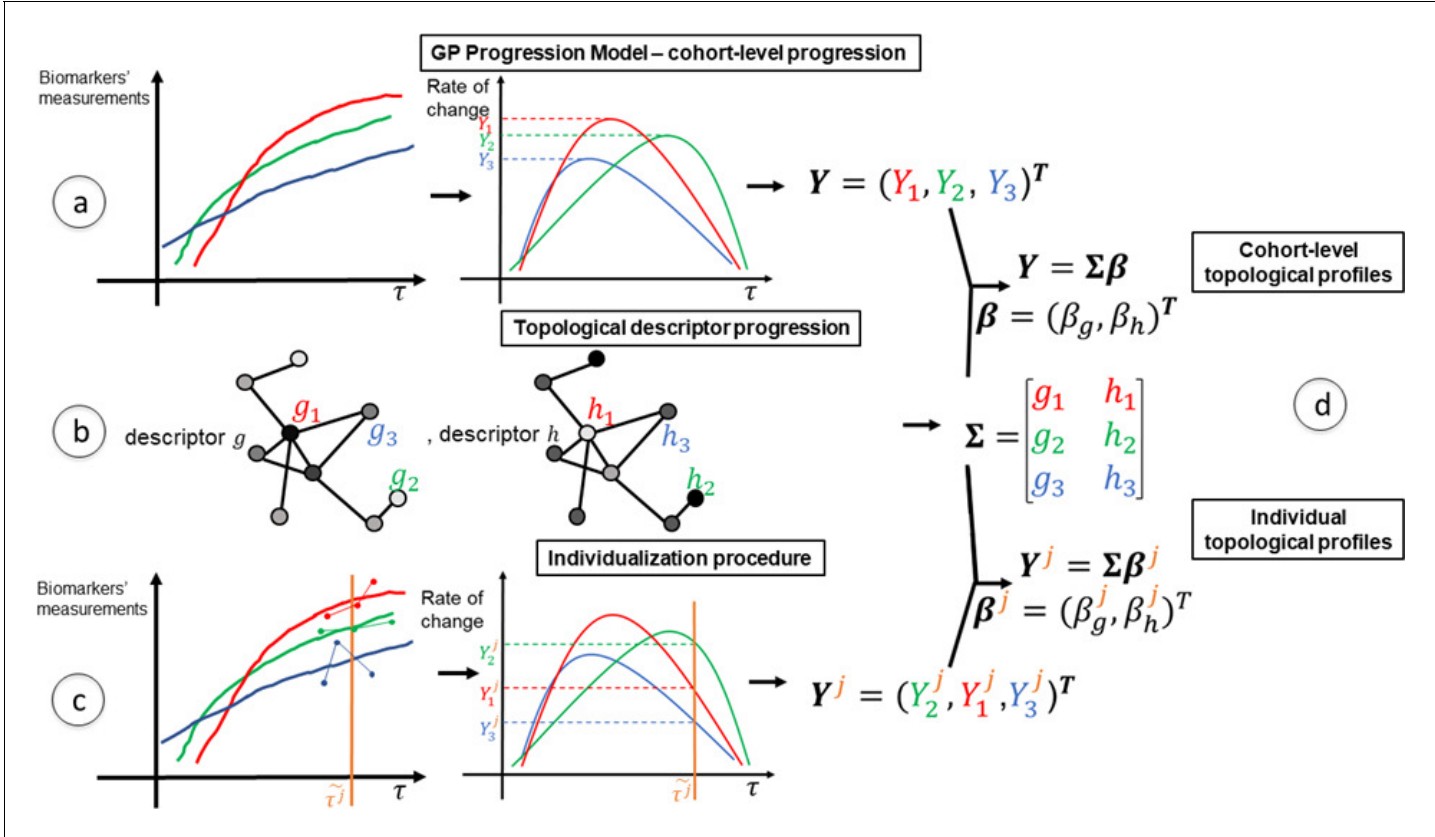

**Figure 5.** A schematic representation of the mathematical modeling of the topological profiling with GP Progression Model. In the example here we have three biomarkers/regions (represented in red ($v = 1$), green ($v = 2$) and blue ($v = 3$)), and two topological descriptors ($g$ and $h$). (a) The GP Progression Model estimates temporal trajectories of biomarkers progression, along the disease time $\tau$. The unique maximum points of the derivatives of the trajectories correspond to their maximal rate of change $Y$. (b) Two topological descriptors are computed for each region from anatomical connectomes. They combine, column-wise, in the matrix $\Sigma$. (c) For each subject $j$, the GP Progression Model estimates a time-reparametrization which shifts individual measurements to the disease time. For each biomarker, the speed of progression of subject $j$ with respect to the cohort progression is the value of the derivative of the biomarker progression at $\tilde{\tau}^j$, which represents the shift of the average age of the subject. (d) Topological profiles are estimated via a linear model relating $\Sigma$ and $Y$ (for the cohort-level topological profile) or $Y^j$ (for the individual profiles).

Formally, if we represent by $\left(X^j(t_1), \ldots, X^j(t_K^j)\right)^T$ the longitudinal measurements of the regional GM volumes associated with each individual $j$ at their $(t_1, \ldots, t_K^j)$ absolute time points (i.e. dates of subject visit or subject's age-at-visit), and consider all measurements obtained at a particular visit of one individual to occur at a particular disease time $\tau_k^j$, where the mapping from absolute time $t$ to disease time $\tau$ is via a subject-specific time reparametrization function, which here is just a simple shift: $t_k^j = \tau_k^j + d^j$, then the observations for subject $j$ at a single time point $t$ (indices omitted but implied) can be modelled as a random sample from the GP Progression Model:

$$\begin{aligned} X^j(t) &:= (X_1^j(t), \ldots, X_V^j(t))^T \\ &= f(t) + \boldsymbol{n}^j(t) + E. \end{aligned} \tag{1}$$

Here, $\boldsymbol{f}(t) = (f_1(t), \ldots, f_V(t))^T$ is the fixed effect function modeling the longitudinal evolution of the $V$ GM volumes, and is modeled as a Gaussian Process; $\boldsymbol{n}^j(t) = \left(n_1^j(t), \ldots, n_V^j(t)\right)^T$ are the individual random effects; and $\boldsymbol{E} = (E_1, \ldots, E_V)^T$ is the observational noise. The model is described in detail in *Desikan et al. (2006)* together with the optimization scheme to recover the probabilistic estimates of the parameters for the fixed effect, the random effect, and the individual time reparametrization

parameters. Identifiability of the model is ensured by enforcing monotonicity on the population-level biomarker trajectories $f_v$.

Fitting the model also provides an estimate of the highest rate of change of each biomarker during the disease progression. That is the maximum of the derivatives of the estimated trajectories $f_v$ along the disease time $\tau$:

$$Y_v = \max_\tau f_v'(\tau), \tag{2}$$

for each biomarker $v \in \{1, \ldots, V\}$. Existence and boundness of $Y_v$ is guaranteed by the finiteness of the time-shifts, which is enforced by the smoothness of the Gaussian Process. Indeed, for each subject $j$, $d^j$ defines the optimal shift of the data point on the temporal time axis. The estimates of these positions must be compatible with the Gaussian Process describing the temporal trajectory (*Figure 2—figure supplement 2, 3 and 4*). Gaussian processes are completely identified by the kernel function (in our case a radial basis function - RBF), which prescribes the shape and smoothness of the interpolating curve via its length-scale $l$ and variance $\sigma$. For this reason, the relative positions (i.e. the time-shifts) of each individual are naturally bounded by the length-scale of the Gaussian process, so the only compatible solutions are those with the time-shifts softly constrained to a finite range determined by $2l$. The monotonicity constraint on $f_v$ guarantees uniqueness of $Y_v$ for each $v$. The model returns uncertainty on $\boldsymbol{f}$, which can be projected to $\boldsymbol{f}'$ and thus $\boldsymbol{Y}$.

## Topological profile estimate

We estimate the topological profile $\beta$ by identifying the unique combination of topological descriptors that best matches the GP Progression. This means estimating the weights $\beta$ from the linear model

$$Y = \Sigma\beta + \varepsilon, \tag{3}$$

where $\boldsymbol{Y} = (Y_1, \ldots, Y_V)^T$; $\Sigma$ collects the values of the metrics $\boldsymbol{g}(v)$ for every region $v$, that is $\Sigma = (\boldsymbol{g}(1), \ldots, \boldsymbol{g}(V))^T$; and $\varepsilon$ is the noise. As a first step we normalize both $\boldsymbol{Y}$ and $\Sigma$ to the range $[0, 1]$. We also enforce non-negativity of $\beta$, so that $\beta_n$ can be precisely interpreted as the weight with which descriptor $n$ contributes to the overall observed pattern of neurodegeneration $\boldsymbol{Y}$. Our problem is then the one of estimating $\beta$ such that:

$$\beta = argmax_{\beta \geq 0} L(\boldsymbol{Y}, \Sigma\beta), \tag{4}$$

where $L$ is the likelihood of the model, which assumes Gaussian noise, so that (4) becomes equivalent to constrained least-squares minimization. We solve the problem via Expectation-Maximization, which, in the positively-constrained least-squared case, has a simple closed form, and becomes an iterative-multiplicative algorithm also known as ISRA (*Daube-Witherspoon and Muehllehner, 1986*; *Dempster et al., 1977*).

## Personalized topological profiles

For each subject, the GP Progression Model estimates a set of time-points $\tau_k^j$ describing the subject's measurements in the new disease time $\tau$ as $t_k^j = \tau_k^j + d^j$, where $d^j$ is the subject-specific shift. If we define the subject temporal position in the disease time as the shifted average age $\tilde{\tau}^j$, where $\tilde{t}^j = \tilde{\tau}^j + d^j$ and $\tilde{t}^j$ is the average age of subject $j$ across age-at-visits, then we can compute the individual speed of progression along the disease time as

$$Y_v^j = f_v'\left(\tilde{\tau}^j\right), \tag{5}$$

for each biomarker $v \in \{1, \ldots, V\}$. These values encode information on the individual rates of change along the disease progression. We can then estimate the individual topological profile $\beta^j$ by identifying the unique combination of topological descriptors that best matches the individual progression as

$$\beta^j = argmax_{\beta^j \geq 0} L(\mathbf{Y}^j, \Sigma\beta^j), \tag{6}$$

where $\mathbf{Y}^j = (Y_1^j, \ldots, Y_V^j)^T$. As in the cohort-level framework, the problem can be solved, for each subject $j$, via ISRA.

## Model selection

In order to quantify the performances of the topological profiles progression against the single-best fitting descriptor in reproducing the observed progression from the GP Progression Model, we compute Aikake Information Criterion (AIC) for each model to balance error scores with model complexity. We assume Gaussian noise and set AIC $= 2N + V\log\left(\frac{\text{RSS}}{V}\right)$, where $N$ is the number of model parameters, $V$ is the number of data points (GM regions), and the RSS is the residual sum of squares between the predicted outcome of the model and the data, that is $\sum_{v=1}^{V}\left((\mathbf{Y}-\Sigma\beta)^2\right)_v$. The single-best fitting descriptor choice has just one parameter, with trans-neuronal spread or proximity spread having one additional parameter for epicenter selection. The topological profiles have one parameter for each descriptor, plus one parameter for epicenter selection, minus one parameter as the weights are normalized, that is as many parameters as the number of the descriptors. The AIC scores are *Supplementary file 1*- Table S4.

## Code availability

Code is available at: https://github.com/sgarbarino/mechanistic-profiles. (*Garbarino, 2019*; copy archived at https://github.com/elifesciences-publications/mechanistic-profiles).

The software (BrainPainter) for coloring brain images as in *Figure 2* is open-source and available at: https://github.com/mrazvan22/brain-coloring.

## Acknowledgements

This project has received funding from the European Union's Horizon 2020 research and innovation programme under grant agreement No. 666992.

SG, NPO, EJV, OC, FB, JMS, MWV, DCA acknowledge funding from the European Union's Horizon 2020 research and innovation programme.

SG acknowledges financial support from the French government managed *by L'Agence Nationale de la Recherche* under *Investissements d'Avenir* UCA[JEDI] (ANR-15-IDEX-01) through the project 'AtroProDem: A data-driven model of mechanistic brain Atrophy Propagation in Dementia'.

NPO acknowledges financial support from The Michael J Fox Foundation for Parkinson's Research, the Alzheimer's Association, Alzheimer's Research UK, and the Weston Brain Institute through 'NetMON: Network Models Of Neurodegeneration' (BAND-15–368107, 11042).

DCA and NPO were also funded from EPSRC grants EP/M020533/1 and EP/J020990/01. DCA, NPO, FB, and JMS are supported by the NIHR UCLH Biomedical Research Centre.

Data collection and sharing for this project was funded by the Alzheimer's Disease Neuroimaging Initiative (ADNI) (National Institutes of Health Grant U01 AG024904). ADNI is funded by the National Institute on Aging, the National Institute of Biomedical Imaging and Bioengineering, and through generous contributions from the following: AbbVie, Alzheimer's Association; Alzheimer's Drug Discovery Foundation; Araclon Biotech; BioClinica, Inc; Biogen; Bristol- Myers Squibb Company; CereSpir, Inc; Eisai Inc; Elan Pharmaceuticals, Inc; Eli Lilly and Company; EuroImmun; F Hoffmann-La Roche Ltd and its affiliated company Genentech, Inc; Fujirebio; GE Healthcare; IXICO Ltd.; Janssen Alzheimer Immunotherapy Research and Development, LLC.; Johnson and Johnson, Pharmaceutical Research and Development LLC.; Lumosity; Lundbeck; Merck and Co., Inc; Meso Scale Diagnostics, LLC.; NeuroRx Research; Neurotrack Technologies; Novartis Pharmaceuticals Corporation; Pfizer Inc; Piramal Imaging; Servier; Takeda Pharmaceutical Company; and Transition Therapeutics. The Canadian Institutes of Health Research is providing funds to support ADNI clinical sites in Canada. Private sector contributions are facilitated by the Foundation for the National Institutes of Health (www.fnih. org). The grantee organization is the Northern California Institute for Research and Education, and the study is coordinated by the Alzheimer's disease Cooperative Study at the University of California,

San Diego. ADNI data are disseminated by the Laboratory for Neuroimaging at the University of Southern California.

The Rotterdam Study is supported by the Erasmus MC and Erasmus University Rotterdam; the Netherlands Organization for Scientific Research; the Netherlands Organization for Health Research and Development; the Research Institute for Diseases in the Elderly; the Netherlands Genomics Initiative; the Ministry of Education, Culture and Science; the Ministry of Health Welfare and Sports; the European Commission; and the Municipality of Rotterdam.

The authors acknowledge The National Institute for Health Research (NIHR) Biomedical Research Centre (BRC) at University College London Hospitals (UCLH) for their support.

## Additional information

### Competing interests

Wiro J Niessen: Founder, scientific lead and shareholder of Quantib BV. The other authors declare that no competing interests exist.

### Funding

| Funder | Grant reference number | Author |
| --- | --- | --- |
| Horizon 2020 Framework Programme | 666992 | Sara Garbarino<br>Marco Lorenzi<br>Neil P Oxtoby<br>Elisabeth J Vinke<br>Olga Ciccarelli<br>Frederik Barkhof<br>Jonathan M Schott<br>Meike W Vernooij<br>Daniel C Alexander |
| Université Côte d'Azur | UCA$^{JEDI}$ ANX 15 IDEX 01 | Sara Garbarino |
| Michael J. Fox Foundation | BAND 15 368107 11042 | Neil P Oxtoby |
| Engineering and Physical Sciences Research Council | EP/M020533/1 | Neil P Oxtoby<br>Daniel C Alexander |
| Engineering and Physical Sciences Research Council | EP/J020990/01 | Neil P Oxtoby |

The funders had no role in study design, data collection and interpretation, or the decision to submit the work for publication.

### Author contributions

Sara Garbarino, Conceptualization, Data curation, Software, Formal analysis, Validation, Visualization, Methodology, Writing - original draft, Writing - review and editing; Marco Lorenzi, Razvan V Marinescu, Software, Methodology; Neil P Oxtoby, Conceptualization, Software, Funding acquisition, Methodology, Writing - original draft, Writing - review and editing; Elisabeth J Vinke, Data curation, Investigation; Arman Eshaghi, Data curation, Investigation, Writing - review and editing; M Arfan Ikram, Wiro J Niessen, Data curation; Olga Ciccarelli, Conceptualization; Frederik Barkhof, Jonathan M Schott, Conceptualization, Writing - review and editing; Meike W Vernooij, Conceptualization, Investigation, Project administration, Writing - review and editing; Daniel C Alexander, Conceptualization, Funding acquisition, Project administration, Writing - review and editing

### Author ORCIDs

Sara Garbarino https://orcid.org/0000-0002-3583-3630
Neil P Oxtoby http://orcid.org/0000-0003-0203-3909
Frederik Barkhof https://orcid.org/0000-0003-3543-3706

### Ethics

The Rotterdam Study has been approved by the Medical Ethics Committee of the Erasmus MC (registration number MEC 02.1015) and by the Dutch Ministry of Health, Welfare and Sport (Population Screening Act WBO, license number 1071272-159521-PG).

### Decision letter and Author response

Decision letter https://doi.org/10.7554/eLife.49298.sa1
Author response https://doi.org/10.7554/eLife.49298.sa2

## Additional files

### Supplementary files

- Supplementary file 1. Supplementary information.
- Transparent reporting form

### Data availability

AD data set from ADNI. ADNI is a public-private partnership. All ADNI data are shared without embargo through the LONI Image and Data Archive (https://ida.loni.usc.edu/login.jsp) a secure research data repository. Interested scientists may obtain access to ADNI imaging, clinical, genomic, and biomarker data for the purposes of scientific investigation, teaching, or planning clinical research studies. Access is contingent on adherence to the ADNI Data Use Agreement. For up-to-date information please see http://adni.loni.usc.edu/wp-content/uploads/how_to_apply/ADNI_DSP_Policy.pdf. PPMS data set from UCLH. Data can be obtained upon request, directed the management team of the data at the Institute of Neurology, UCL: uclh.qsmsc@nhs.net. HA data set from the Rotterdam Study. Data can be obtained upon request. Requests should be directed towards the management team of the Rotterdam Study (secretariat.epi@erasmusmc.nl), which has a protocol for approving data requests. Because of restrictions based on privacy regulations and informed consent of the participants, data cannot be made freely available in a public repository. The Rotterdam Study has been approved by the Medical Ethics Committee of the Erasmus MC (registration number MEC 02.1015) and by the Dutch Ministry of Health, Welfare and Sport (Population Screening Act WBO, license number 1071272-159521-PG). The Rotterdam Study has been entered into the Netherlands National Trial Register (NTR; www.trialregister.nl) and into the WHO International Clinical Trials Registry Platform (ICTRP; www.who.int/ictrp/network/primary/en/) under shared catalogue number NTR6831. All participants provided written informed consent to participate in the study and to have their information obtained from treating physicians. HCP data are from the Human Connectome Project. Open Access Data (all imaging data and most of the behavioral data) is available to those who register an account at ConnectomeDB and agree to the Open Access Data Use Terms. This includes agreement to comply with institutional rules and regulations. For up-to-date information please see https://www.humanconnectome.org/study/hcp-young-adult/data-use-terms.

The following datasets were generated:

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
