## [Decision Letter]

**Acceptance summary:**

This study aimed to characterize the temporal evolution of different neurological disorders in terms of their topological profile – a combination of graph-theoretical descriptors (e.g., centrality, segregation) that together best describe the progression of pathology. The study clearly shows the advantage of looking at a combination of topological features, rather than a single descriptor; and investigating disease progression longitudinally, rather than relying on end-stage data. Moreover, this work sets the stage for potentially improving the sensitivity of clinical diagnosis. As such, this work may be of significant interest to both fundamental researchers interested in disease mechanisms and clinicians aiming to use state-of-the-art methods to improve diagnostic success.

**Decision letter after peer review:**

[Editors’ note: this article was originally rejected after discussions between the reviewers, but the authors were invited to resubmit after an appeal against the decision.]

Thank you for submitting your work entitled "Differences in topological progression profile among neurodegenerative diseases from imaging data" for consideration by *eLife*. Your article has been reviewed by a Reviewing Editor and a Senior Editor, a Reviewing Editor, and two reviewers. The reviewers have opted to remain anonymous.

We recognize that the paper introduces the concept of "topological profiles", and using this novel method, can distinguish between neurological conditions. Unfortunately, the reviewers have raised concerns with the manuscript both in methodology and scientific advance. After discussion between with the editors and reviewers, the decision is to reject the paper without possibility of reconsideration at *eLife. eLife* is focused on publishing the most exciting and impactful research, but in this instance, the study fell short of our bar for publication.

*Reviewer #1:*

In this study by Garbarino et al., the authors investigated the topological progression profiles of Alzheimer's disease (AD), primary progressive MS (PPMS) and normal aging. They use the term "topological profile" which they define as the combination of topological descriptors showing the pathology propagation of a particular disease.

By investigating the relationship between topology of brain network connectivity and pattern of pathology, they aim to understand underlying mechanisms of propagation in AD and PPMS.

The neurodegeneration profile was found different in AD and PPMS as expected. The neurodegeneration profile in AD and PPMS were also different from the community-dwelling aging individuals (HA), however since this was a longitudinal study, what is the contribution of the aging process in AD and PPMS groups? Can the topological profile detect the influence of aging in the AD and PPMS groups? What was the mean age in each group? Was any age matching used? What was the time between each longitudinal scan?

Please discuss why the PPMS outliers showed no significant differences compared with the rest of the PPMS group.

If a PPMS patient is presented with dementia, although clinical features usually point out the origin, can the topological profile also distinguish whether dementia is due to MS or AD? Providing a case example may help.

Is the topological profile method applicable to other phenotypes of MS? Progressive MS (primary or secondary) is essentially the same except for preceding relapse(s), but there are more secondary progressive MS patients and therefore any clinical application would be more meaningful. Please discuss.

As neurodegeneration likely starts in the relapsing phase of MS, all the way back to asymptomatic phase of MS (radiologically isolated syndrome – RIS); please discuss the potential use of the "topological profile" method in RIS.

What is the clinical application of this method, how does this approach change the clinical evaluation of patients?

How does this method help understand the underlying mechanisms? Please discuss.

*Reviewer #2:*

This work aims at better understanding the longitudinal progression of gray matter atrophy in the brain in disease and aging. Specifically, it aims at understanding the topological organization of grey matter atrophy as a function of the disease progression, in parallel, for multiple diseases, including Dementia and Multiple Sclerosis. Three disjoint datasets (or cohorts) are used.

The methodology proceeds in four steps: (1a) computation of ROI descriptors for each ROI and each image. (1b) computation of graph-based (or topological) features for each ROI. (2) Computation of grey matter atrophy trajectories for each ROI and each dataset using a disease Progression Model (DPM). (3) Regression of the output of the DPM onto the graph-based features and (4) Statistical analysis of this regression.

In my view, the approach is of great interest. However, the analysis in steps (3) and (4) is lacking important details without which it is difficult to assess the validity and replicability of the approach. Specifically,

ν

A) in equation (2), what is the range of values of τ that is used? In the DPM, τ is not bounded as I understand, thus Y_ν_ might not even exist. In figure 2, in the 4 columns related to AD, how do you explain that the topological profiles are different while the left-hand side of (2) does not depend on τ. The same question applies to the columns related to PPMS and HA.

B) How much of the variance is explained in (3)? This is important because at this point it is not clear if the topological features, aside from the constant term, have any significant explanatory power. On a similar note, the null hypothesis β = 0 in Table 1 should be replaced by a less stringent baseline, e.g. β = 0 aside of the term associated with the constant progression. Other confounding, e.g. volume in healthy subjects could also be investigated.

In summary, the methodology is not rigorous enough to assess the validity and reproducibility of the findings in my view.

---

## [Author Response]

We recognize that the paper introduces the concept of "topological profiles", and using this novel method, can distinguish between neurological conditions.

1) To be clear, the primary message is not about distinguishing different conditions, per se. It is that the topological profiles explain image data sets much better than individual topological components. Individual topological components have been widely used in high profile neuroscience publications to inform on putative disease propagation mechanisms, but, as we show here, do not explain observed variation in the data well. This novel observation suggests that multiple concurrent mechanisms of disease spread contribute to observed pathology patterns and that distinct combinations characterise different neurological conditions. This novel insight is the primary contribution of the paper. The ideas do also have potential clinical utility by providing integrated biomarkers for differential early diagnosis, but that is a secondary thought to be explored fully in future work.

Unfortunately, the reviewers have raised concerns with the manuscript both in methodology and scientific advance.

2) We do not believe any substantial methodological concerns are raised, as we discuss below.

Moreover, reviewer 2 describes the primary scientific advances as “of great interest”. Reviewer 1 does raise questions about the potential for clinical application, which we address in detail below, but we would like to emphasise that the primary aim of this manuscript is biological insight into disease process rather than providing a clinical tool.

Reviewer #1:In this study by Garbarino et al., the authors investigated the topological progression profiles of Alzheimer's disease (AD), primary progressive MS (PPMS) and normal aging. They use the term "topological profile" which they define as the combination of topological descriptors showing the pathology propagation of a particular disease.By investigating the relationship between topology of brain network connectivity and pattern of pathology, they aim to understand underlying mechanisms of propagation in AD and PPMS.The neurodegeneration profile was found different in AD and PPMS as expected. The neurodegeneration profile in AD and PPMS were also different from the community-dwelling aging individuals (HA), however since this was a longitudinal study, what is the contribution of the aging process in AD and PPMS groups? Can the topological profile detect the influence of aging in the AD and PPMS groups?

3) We thank the reviewer for his/her comment. As stated in subsection “Data Preprocessing”, we regress out the effects of ageing in the AD and PPMS cohorts to minimise the effects of the ageing process on disease models. Nevertheless, some component of ageing may appear in the models for both; that is why we include the HA models for comparison. We have now explained that logic more clearly in the Discussion section:

“Further, we note that we have regressed out the effects of ageing in the AD and PPMS cohort in order to minimize the effects of the ageing process on disease models. Nevertheless, some components of ageing might still appear in the model for both, which is one key reason for including the HA model for comparison.”

What was the mean age in each group? Was any age matching used? What was the time between each longitudinal scan?

4) All this information was in the original Supplementary material. It remains in Supplementary file Table S6, Table S7, Table S8 and Figure 3—figure supplement 1. However, we have now added these important facts to the main body of the paper, in the Subsection “Data description”:

“Participants-AD […] the mean (std) age of the cohort is 73.9(7.2) years; […] the average (std) time between scans is 2.4(1.8) years.”

“Participants-PPMS […] the mean (std) age of the cohort is 41.6(10.2) years; […] the average (std) time between scans is 4.0(1.5) years.”

“Participants-HA […] the mean (std) age of the cohort is 64.8(10.8) years; […] the average (std) time between scans is 5.3(1.1) years.”

No age-matching criterion was used (full details on data processing in the subsection “Data Preprocessing”). Since we aim to use volume loss across the full time-course of disease (or ageing process) to gain insight on underlying mechanisms, we use the largest possible age-range for each cohort. Nevertheless, we understand the reviewer’s concern that perhaps differences in age-range might explain differences in topological profile. To address this, we have added an experiment that computes topological profiles on subsets of the HA cohort age-matched with the AD and MS cohorts. Both produce quantitatively similar topological profiles to those we see on the full cohort, as described in the Revised Manuscript subsection “Distinct topological profiles for each neurological condition” (and in the revised Supplementary file Table S2):

“Supplementary Information (Table S2) shows the topological profiles for two subsets of the HA cohort, age-matched with the AD and PPMS cohorts, as compared to topological profiles for the whole HA cohort; results remain consistent.”

Please discuss why the PPMS outliers showed no significant differences compared with the rest of the PPMS group.If a PPMS patient is presented with dementia, although clinical features usually point out the origin, can the topological profile also distinguish whether dementia is due to MS or AD? Providing a case example may help.

5) We would have expected PPMS subjects whose topological profile is closer to the AD

(respectively HA) cohort profile to show features that align them more closely to the AD (respectively HA) cohort. We believe we do not observe statistical significance for two main reasons: (i) sample size: our method estimated only 1 subject’s topological profile to be closer to the AD profile, and 13 to HA; and (ii) available clinical markers: just age, gender, disease duration and EDSS were collected on the PPMS cohort. Future investigation on larger cohorts, as well as with a broader class of clinical features, could help to clarify whether the differences we observe in terms of topological profiles are spurious effects related to small sample size, or instead carry information on the clinical status of the outliers that is not identifiable using the available features. We have now commented on this in the subsection “Degree of alignment of individual profiles with cohort profiles informs on patient status”:

“We found no significant difference in clinical features between PPMS outliers, tending towards either AD or HA, compared with the rest of the PPMS group: topological profiles seem to identify differences among PPMS subjects that are invisible to the available demographic and clinical features (age, gender, EDSS and diseases duration). However, sample sizes are low (N=1 PPMS-AD outlier; N=13 PPMS-HA outliers). Further investigation on a larger cohort of progressive MS subjects, as well as with a broader class of clinical features, may clarify whether the variation we observe in topological profiles arise from spurious effects, or are genuinely informative on the clinical status of the outliers beyond what the available clinical features can identify.”

Regarding the second concern raised by the reviewer, the key idea of our proposed method is that two diseases have distinguishable topological profiles. For this reason, we believe our method would be able to distinguish between subjects with dementia arising from, for instance, MS-induced neurodegeneration and subjects whose dementia arose from AD. Nevertheless, we note that patients in our PPMS cohort were diagnosed with “early” PPMS (they were recruited to this study within 5 years of diagnosis). Although a detailed cognitive assessment was not performed, in such a population of early PPMS it is very unlikely to have patients with PPMS who present with dementia, as for instance outlined in Chiaravallotti and DeLuca (2008). For this reason, we could not really test the hypothesis suggested by the reviewer, but it is something that would definitely be of interest for future work. We thank the reviewer for suggesting such an analysis. We have now commented in the subsection “Degree of alignment of individual profiles with cohort profiles informs on patient status”:

“For example, future studies of older and longer-term PPMS patients could test the hypothesis that individual topological profiles may inform on co-morbidities, e.g. for an MS-patient who shows signs of dementia, their profile position with respect to the AD cohort profile may inform on whether the dementia comes from MS-related neurodegeneration or AD comorbidity. However, we cannot test this hypothesis here, as our PPMS cohort are “early” PPMS (they were recruited to this study within 5 years of diagnosis), and no subject reported dementia as would be expected^76^ although a detailed cognitive assessment was not performed, this agrees.”

Is the topological profile method applicable to other phenotypes of MS? Progressive MS (primary or secondary) is essentially the same except for preceding relapse(s), but there are more secondary progressive MS patients and therefore any clinical application would be more meaningful. Please discuss.As neurodegeneration likely starts in the relapsing phase of MS, all the way back to asymptomatic phase of MS (radiologically isolated syndrome – RIS); please discuss the potential use of the "topological profile" method in RIS.

6) Yes, in principle, the method is readily applicable to a range of diseases or disease phenotypes (as stated in subsection “Future wWork”). However, we reiterate a point from the paper (subsection “Future work”), that our approach assumes some degree of consistency in topological profile within cohorts. We focus on PPMS as a more degenerative phenotype that is relatively homogeneous compared to other groups (e.g. SPMS). We do intend to look at subtle differences among subtypes in future work (subsection “Future work”), but that likely needs better identification of distinct subgroups than CIS/RRMS/PPMS/SPMS/etc., e.g. using techniques like SuStaIn (Young, 2018; Eshaghi, 2019). We have expanded our discussion of this in the subsection “Future work”:

“Indeed, we believe application to broader cohorts, as for instance generic progressive MS including both primary and secondary (which would be more numerous) would require better identification of distinct subgroups than the traditional MS classifications (clinically isolated syndrome, relapsing remitting MS, and primary and secondary progressive MS), and methods such as SuStain^87^ would offer a way to identify within-cohort subtypes for better topological profile estimation.”

Application to RIS subjects is of course of great interest for future work. However, further to the question of heterogeneity above, we add here that it is the whole disease progression process that informs topological profiles. Thus, it would not make sense to construct a profile for RIS – the early stage of a longer process – in isolation. We believe instead that the key opportunity for future clinical utility in MS is to identify distinct trajectories of pathology accumulation, as in (Eshaghi, 2019), use the topological profile to reveal potential differences in underlying processes, and then identify which profile each individual RIS subject aligns with best thereby informing treatment and care choices. We have added a comment on this in the Revised Manuscript, subsection “Future work”:

“As concerns applications to prodromal phases of diseases – an example being radiologically isolated syndrome for MS – future work will be devoted to build models of the whole neurodegeneration process within specific diseases and their subtypes defined by distinct trajectories of pathological change, which might be used for instance for discriminating subjects that convert to degeneration from others that do not, thereby informing treatment and care choices.”

What is the clinical application of this method, how does this approach change the clinical evaluation of patients?

7) We emphasise again that the intention here is to enhance disease understanding rather than provide a front-line clinical tool. Impact is more likely to come from inspiring future treatment development than informing decisions on individual patients. Nevertheless, the results we provide suggest potential utility for early detection in the clinic and for clinical trials. For example, AD-like HA outliers may be considered at risk for AD and decisions could be informed by this, including patient management (more frequent visits to monitor the patient) and recruitment into clinical studies/trials. We have added a comment on this in the subsection “Future work”:

“With further development and experiments, topological profiles may also provide useful information in the clinic. For example, AD-like HA outliers may be considered at risk for AD and decisions could be informed by this, including patient management (more frequent visits to monitor the patient) and recruitment into clinical studies/trials.”

How does this method help understand the underlying mechanisms? Please discuss.

8) In brief, the idea is that different topological components represent different mechanisms of pathology spread so that the topological profile suggests a combination of mechanisms characteristic of a particular disease. […] We have expanded this discussion to help the reader understand the relationship more precisely in the Introduction and Discussion section:

“Thus comparing patterns of pathology predicted by these different descriptors with those observed in patient cohorts gives clues to which corresponding mechanisms are at play.”

“As discussed in the introduction, one key implication here (as in previous literature ^3,20,67,68^) is that graph-theoretic measures may associate with different mechanisms of disease spread in neural networks. […] While we draw on these potential links in the following sections, it is important to note that the links between the mathematical abstraction and the biological mechanisms are simplistic, and graphtheory metrics do not fully capture or explain the range of potential spreading mechanisms.”

Reviewer #2:This work aims at better understanding the longitudinal progression of gray matter atrophy in the brain in disease and aging. Specifically, it aims at understanding the topological organization of grey matter atrophy as a function of the disease progression, in parallel, for multiple diseases, including Dementia and Multiple Sclerosis. Three disjoint datasets (or cohorts) are used.The methodology proceeds in four steps: (1a) computation of ROI descriptors for each ROI and each image. (1b) computation of graph-based (or topological) features for each ROI. (2) Computation of grey matter atrophy trajectories for each ROI and each dataset using a disease Progression Model (DPM). (3) Regression of the output of the DPM onto the graph-based features and (4) Statistical analysis of this regression.In my view, the approach is of great interest. However, the analysis in steps (3) and (4) is lacking important details without which it is difficult to assess the validity and replicability of the approach. Specifically,A) in equation (2), what is the range of values of τ that is used? in the DPM, τ is not bounded as I understand, thus Y_ν_ might not even exist. In figure 2, in the 4 columns related to AD, how do you explain that the topological profiles are different while the left-hand side of (2) does not depend on τ. The same question applies to the columns related to PPMS and HA.

9) Although we impose no range of values on the time-shifts *d^j^* (where tkj=τkj+dj for each time point *k* each individual *j*) a-priori, the smoothness of the Gaussian process forces each *d^j^* to be finite and thus *Y_ν_* is guaranteed to exist. The estimate of each *d^j^* must be compatible with the interpolating curve (in our case a monotonic Gaussian process) describing the temporal trajectory (Figure 2—figure supplement 2, Figure 2—figure supplement 3 and Figure 2—figure supplement 4). Gaussian processes are completely identified by the kernel function (in our case a radial basis function – RBF), which prescribes the shape and smoothness of the interpolating curve via its length-scale *l* and variance σ. For this reason, the relative positions (i.e. the time-shift) of each individual are naturally bounded by the length-scale of the Gaussian process. The length-scale automatically regularises the estimation of the time shifts, which cannot diverge towards infinity without compromising the smoothness and the shape of the GP model. Specifically, consider two acquisition times *t*_1_ and *t*_2_ for two distinct subjects. Suppose *t*_2_ to be fixed (we do not shift it), while the shift for *t*_1_ is *d*_1_ ≠ 0. Then, the joint probability distribution of the two data points, governed by the RBF kernel of the GP, is

pt1+d1-t2d1=σ2et1+d1-t222l

As d1→∞,t1+d1-t2 becomes large compared to 2*l*, the associated joint probability goes to zero (this holds for the joint distribution of any set of points). So, the only compatible solutions are those with the time-shifts softly constrained to a finite range determined by 2*l*. We added a comment on this to the Revised Manuscript, subsection “GP progression model”:

“Existence and boundness of Y_ν_ is guaranteed by the finiteness of the time-shifts, which is enforced by the smoothness of the Gaussian Process. Indeed, for each subject *j*, *d^j^* defines the optimal shift of the data point on the temporal time axis. The estimates of these positions must be compatible with the Gaussian Process describing the temporal trajectory (Figure 2—figure supplement 2, Figure 2—figure supplement 3 and Figure 2—figure supplement 4). Gaussian processes are completely identified by the kernel function (in our case a radial basis function – RBF), which prescribes the shape and smoothness of the interpolating curve via its *lengthscale l* and variance σ. For this reason, the relative positions (i.e. the time-shifts) of each individual are naturally bounded by the length-scale of the Gaussian process, so the only compatible solutions are those with the time-shifts softly constrained to a finite range determined by 2*l*.”

Regarding interpretation of Figure 2: while the rates are time-independent, they vary spatially so that the overall pattern is time dependent. We now point this out explicitly in the Revised Manuscript, Results section:

“Each topological profile specifies the rate of pathology accumulation in each area. So, while the rates are time-independent, they vary spatially so that the pattern itself is time dependent.”

B) How much of the variance is explained in (3)? This is important because at this point it is not clear if the topological features, aside from the constant term, have any significant explanatory power. On a similar note, the null hypothesis β = 0 in Table 1 should be replaced by a less stringent baseline, e.g. β = 0 aside of the term associated with the constant progression. Other confounding, e.g. volume in healthy subjects could also be investigated.

10) First, the topological profiles do have substantial explanatory power. For instance, in the AD cohort, the topological features explain 82% of the variance, with the constant term explaining just 6%; this contrasts with 51% explained by the best individual topological feature. We have now added this important information in the Revised Manuscript, subsection “Topological profiles match disease progression better than any single descriptor– thank you for this idea:

“Further, we note that topological profiles explain variance in the data better than the best-fitting single descriptors. Indeed, in the AD cohort, the topological profile explains 82% of the variance, with the constant term explaining just 6%, in contrast to 51% explained by network proximity. In the PPMS cohort, the topological profile explains 83%, the constant term 7%, and inverse clustering 25%. Similarly, in the HA cohort the topological profile explains 88%, the constant term 16%, and cortical proximity 64%.”

Second, we agree the comparison against the alternative baseline is more useful to show, and is now in the revised manuscript (Revised Manuscript, p4, lines 146-150 and revised Table 1): it does not change any conclusions. Again, thanks for the idea.

*The reported p-values and effect sizes (in braces) are relative to the null hypothesis of β* = 0 *aside from the term associated with the constant progression, computed via permutation testing and Bonferroni-corrected for multiple comparison across the set of network metrics. All p-values were found <0.01 apart from the inverse degree for AD, for which p=0.048.*

Finally, total brain volume is factored out, so is not a confound. We also factor out gender and age (age not factored out of the HA cohort), as declared in the subsection “Data Preprocessing”. If the reviewer has other factors to suggest we would be happy to discuss and include them in the processing.

In summary, the methodology is not rigorous enough to assess the validity and reproducibility of the findings in my view.

11) We hope the additional data and explanations we now include convey the validity and reproducibility.

First, the method builds on well-established techniques previously published in respected journals and applied in a variety of contexts (see [Lorenzi et al., 2017; Lorenzi et al., 2018; Abi Nader et al., 2019; Garbarino et al., 2019; Lorenzi et al., 2019] for the GP Progression Model literature, and [Seeley et al., 2009; Zhou et al., 2012; Raj et al., 2012; Gardner et al., 2013; Raj et al., 2015; Fornito et al., 2015; Cope et al., 2018] for graph-theory metrics approach for neurodegeneration modelling in neurodegenerative diseases). As for validity of the findings, we have now added an experiment confirming that features extracted from topological profiles, specifically the individual distances from the cohort-profile, correlate negatively with individual cohort-specific clinical/demographic features. Specifically, MMSE for AD (R=0.11, p<0.01); EDSS for PPMS (R=0.68, p=0.07); and age for HA (R=0.32, p<0.01). This is now in the subsection “Features derived from the topological profile correlate with clinical features” and the Discussion section:

“Finally and more generally, we analysed features extracted from topological profiles, in particular, the distance of individual profiles from the cohort profile, and show that they correlate negatively with individual cohort-specific clinical features. Specifically, MMSE for the AD cohort (R=0.11, p<0.01); EDSS for PPMS (R=0.68, p=0.07); and age for HA (R=0.32, p<0.01).

Further, we retrieved significant correlation between features of the topological profiles and individual clinical or demographic features, suggesting potential clinical utility of the topological profile.”

As for reproducibility of results, as stated in the original manuscript, subsection “Personalized topological profiles”, we have released code github.com/sgarbarino/mechanistic-profiles openly together with a dummy dataset mimicking ADNI (for testing), while processed unrecognizable cohort data sets can be provided upon request. For reproducibility of findings, the paper shows (Table 1 and Figure 3) topological profiles under cross-validation by bootstrapping (n = 100), which consistently support the key conclusions.